# NEURAL PROBABILISTIC LOGIC PROGRAMMING IN DISCRETE-CONTINUOUS DOMAINS

## ABSTRACT

Neural-symbolic AI (NeSy) methods allow neural networks to exploit symbolic background knowledge. NeSy has been shown to aid learning in the limited data regime and to facilitate inference on out-of-distribution data. Neural probabilistic logic programming (NPLP) is a popular NeSy approach that integrates probabilistic models with neural networks and logic programming. A major limitation of current NPLP systems, such as DeepProbLog, is their restriction to discrete and finite probability distributions, e.g., binary random variables. To overcome this limitation, we introduce DeepSeaProbLog, an NPLP language that supports discrete and continuous random variables on (possibly) infinite and even uncountable domains. Our main contributions are 1) the introduction of DeepSeaProbLog and its semantics, 2) an implementation of DeepSeaProbLog that supports inference and gradient-based learning, and 3) an experimental evaluation of our approach.

## 1 INTRODUCTION

Neural-symbolic AI (NeSy) (Garcez et al., 2002; De Raedt et al., 2021) focuses on the integration of symbolic and neural methods. The advantage of NeSy methods is that they combine the reasoning power of logical representations with the learning capabilities of neural networks. Such methods have been shown to converge faster during learning and to be more robust (Rocktäschel and Riedel, 2017; Xu et al., 2018; Evans and Grefenstette, 2018). The challenge of NeSy lies in combining discrete symbols with continuous and differentiable neural representations. So far this has been accomplished by interpreting the outputs of neural networks as the weights of Boolean variables. These weights can either be given a fuzzy semantics (Donadello et al., 2017; Diligenti et al., 2017) or a probabilistic semantics (Manhaeve et al., 2018; Yang et al., 2020). The latter is also used in neural probabilistic logic programming (NPLP) (De Raedt et al., 2019), where neural networks parametrize probabilistic logic programs.

A shortcoming of traditional probabilistic NeSy approaches is that they fail to capture models that integrate continuous random variables and neural networks – a feature that has already been achieved with mixture density networks (Bishop, 1994) and also more generally within a deep probabilistic programming (DPP) setting (Tran et al., 2017; Bingham et al., 2019). Despite the expressiveness of these methods, they have so far focused on efficient probabilistic inference in continuous domains, e.g., Hamiltonian Monte Carlo or variational inference. It is unclear whether they can be generalised to enable logical and relational reasoning. This exposes a gap between DPP and NeSy as reasoning is, after all, a fundamental component of the latter. We close the DPP-NeSy gap by introducing DeepSeaProbLog[1]. DeepSeaProbLog is an NPLP language with support for discrete-continuous random variables that retains logical and relational reasoning capabilities. More concretely, we allow for neural networks to parameterize arbitrary and differentiable probability distributions. We achieve this using the reparameterization trick (Ruiz et al., 2016) and continuous relaxations (Petersen et al., 2021). This stands in contrast to DeepProbLog (Manhaeve et al., 2018) where only finite categorical distributions are supported.

Our main contributions are (1) the well-defined probabilistic semantics of DeepSeaProbLog, a differentiable discrete-continuous NPLP language, (2) an implementation of inference and gradient-based learning algorithms, and (3) an experimental evaluation showing the necessity of discrete-continuous reasoning and the efficacy of our approach.

---

[1] 'Sea' stands for the letter C, as in **c**ontinuous random variable.

## 2 LOGIC PROGRAMMING CONCEPTS

A term `t` is either a constant `c`, a variable `V` or a structured term of the form `f(t₁,...,t_K)`, where `f` is a functor and each `t_i` is a term. Atoms are expressions of the form `q(t₁,...,t_K)`. Here, `q/K` is a predicate of arity $K$ and each `t_i` is a term. A literal is an atom or the negation of an atom `¬q(t₁,...,t_K)`. A definite clause (also called a rule) is an expression of the form `h:- b₁,...,b_K` where `h` is an atom and each `b_i` is a literal. Within the context of a rule, `h` is called the head and the conjunction of `b_i`'s is referred to as the body of the rule. Rules with an empty body are called facts. A logic program is a finite set of definite clauses. If an expression does not contain any variables, it is called ground. Ground expressions are obtained from non-ground ones by means of substitution. A substitution $\theta = \{V_1 = t_1, \ldots, V_K = t_K\}$ is a mapping from variables $V_i$ to terms $t_i$. Applying a substitution $\theta$ to an expression `e` (denoted `e`$\theta$) replaces each occurrence of $V_i$ in `e` with the corresponding $t_i$.

While *pure* Prolog (or definite clause logic) is defined using the concepts above, practical implementations of Prolog extend definite clause logic with an external arithmetic engine (Sterling and Shapiro, 1994, Section 8). Such engines enable the use of system specific routines in order to handle numeric data efficiently. Analogous to standard terms in definite clause logic, as defined above, we introduce numeric terms. A numeric term $n_i$ is either a numeric constant (a real, an integer, a float, etc.), a numeric variable $N_i$, or a numerical functional term, which is an expression of the form $\varphi(\text{n}_1,\ldots,\text{n}_K)$ where $\varphi$ is an externally defined numerical function. The difference between a standard logical term and a numerical term is that *ground* numerical terms are evaluated and yield a numeric constant. For instance, if `add` is a function, then `add(3, add(5, 0))` evaluates to the numerical constant `8`.

Lastly, numeric constants can be compared to each other using a built-in binary comparison operator $\bowtie \in \{<, =<, >, >=, =:=, =\backslash=\}$. Here we use Prolog syntax to write comparison operators, which correspond to $\{<, \leq, >, \geq, =, \neq\}$ in standard mathematical notation. Comparison operators appear in the body of a rule, have two arguments, and are generally written as $\varphi_l(\text{n}_{l,1},\ldots,\text{n}_{n,K}) \bowtie \varphi_r(\text{n}_{r,1},\ldots,\text{n}_{r,K})$. They evaluate their left and right side and subsequently compare the results, assuming everything is ground. If the stated comparison holds, the comparison is interpreted by the logic program as true, else as false.

## 3 DEEPSEAPROBLOG

### 3.1 SYNTAX

While facts in pure Prolog are deterministically true, in probabilistic logic programs they are annotated with the probability with which they are true. These are the so-called probabilistic facts (De Raedt et al., 2007). When working in discrete-continuous domains, we need to use the more general concept of distributional facts (Zuidberg Dos Martires, 2020), inspired by the distributional clauses of Gutmann et al. (2011).

**Definition 3.1** (Distributional fact). Distributional facts are expressions of the form `x ~ distribution(n₁,...,n_K)`, where `x` denotes a term, the `n_i`'s are numerical terms and `distribution` expresses the probability distribution according to which `x` is distributed.

The meaning of a distributional fact is that all ground instances `x`$\theta$ serve as random variables that are distributed according to `distribution(n₁,...,n_K)`$\theta$. All variables appearing on the right-hand side of a distributional fact must also appear on its left-hand side.

**Definition 3.2** (Neural distributional fact). A neural distributional fact (NDF) is a distributional fact in which a subset $\{\text{f}_j\}_{j=1}^L \subseteq \{\text{n}_i\}_{i=1}^K$ of the set of numerical terms in the distributional fact is implemented by neural networks that depend on a set of neural parameters $\{\boldsymbol{\lambda}_j\}_{j=1}^L$.

**Example 3.1** (DeepSeaProbLog program). Consider the DeepSeaProbLog program below where `humid(Data)` denotes a Bernoulli random variable that takes the value 1 with probability $p$ given by the output of a neural network `humidity_detector`. `temp(Data)` denotes a normally distributed variable whose parameters are predicted by a network `temperature_predictor`. The program further contains two rules that deduce whether we have good weather or not.

```
humid(Data) ~ bernoulli(humidity_detector(Data)).
temp(Data) ~ normal(temperature_predictor(Data)).
snowy_weather ~ beta(2, 7). sunny_weather ~ beta(5, 3).

good_weather(Data, Degree) :-
    humid(Data) =:= 1, temp(Data) < 0, snowy_weather < Degree.
good_weather(Data, Degree) :-
    humid(Data) =:= 0, temp(Data) > 15, sunny_weather > Degree.

query(good_weather(data1, degree1)).
```

The query atom at the end declares the probability of the atom we would like to compute and also tells us which ground term to replace the logic variables with.

Notice that the random variables `humid(Data)` and `temp(Data)` appear in the body of the logical rule with comparison operators. So far, these comparisons were interpreted by the logic program as deterministically true or false. In the probabilistic setting, the truth value of the comparison depends on the value of the random variable and is thus random itself. Furthermore, to obtain well-defined probability distributions, we need to restrict these comparison operators to being Lebesgue-measurable.

**Definition 3.3.** (Probabilistic comparison formula) A *probabilistic comparison formula* (PCF) is an expression of the form $(g(\boldsymbol{x}) \bowtie 0)$, where $g$ is a function applied to the set of random variables $\boldsymbol{x}$ and $\bowtie \in \{<, =<, >, >=, =:=, =\backslash=\}$ is a *comparison operator*. A valid PCF defines a *measurable* set as $\{\boldsymbol{x} \mid g(\boldsymbol{x}) \bowtie 0\}$.

Note that in Definition 3.3, we write the general form of a PCF with a $0$ on the right-hand side. This is without loss of generality, as we can always obtain this form by subtracting the right hand-side from both sides of the relation. With the definitions of NDFs and PCFs, a DeepSeaProbLog program can now be formally defined.

**Definition 3.4** (DeepSeaProbLog program). A DeepSeaProbLog program consists of a finite set of NDFs $\mathcal{F}_D$ (defining random variables), a finite set $\mathcal{C}_M$ of valid PCFs and a set of logical rules $\mathcal{R}_L$ that can use any of those valid PCFs in their bodies.

DeepSeaProbLog generalises a range of existing PLP languages. For instance, if we were to remove the distributional facts on `temp(Data)`, `snowy_weather` and `sunny_weather` and all the PCFs using them, we would obtain a DeepProbLog program (Manhaeve et al., 2018). If we additionally replace the neural network in `humid` with a fixed probability `p`, we end up with a probabilistic logic program (De Raedt et al., 2007). Alternatively, replacing the constant probability `p` by a constant `1` yields a non-probabilistic Prolog program. Similarly, considering all rules and facts in Example 3.1 but replacing the neural parameters of the normal distribution with numeric constants results in a Distributional Clauses program (Gutmann et al., 2011). We discuss the connection of DeepSeaProbLog to these related languages further in Appendix A, where we also formally state and prove the reduction from DeepSeaProbLog to DeepProbLog.

### 3.2 SEMANTICS

DeepSeaProbLog programs are used to answer probabilistic queries of ground atoms, i.e. to compute the probability with which a ground atom `q` is satisfied. The probability itself follows from the semantics of the DeepSeaProbLog program. As is custom in (probabilistic) logic programs, we will define the semantics of DeepSeaProbLog with regard to ground programs.

We will assume that the set of distributional facts $\mathcal{F}_D$ is *valid*, which means that the random variables it defines must all be unique, i.e., each distributional fact must define a different random variable. Notice also that the resulting ground distributional facts will contain the inputs to the neural functions. In a sense, a DeepSeaProbLog program is conditioned on these neural network inputs.

To define the semantics of ground DeepSeaProbLog programs, we first introduce the possible worlds over the PCFs. Every subset $C_M$ of a set of PCFs $\mathcal{C}_M$ defines a possible world $\omega_{C_M} = \{C_M \cup h\theta \mid \mathcal{R}_L \cup C_M \models h\theta \text{ and } h\theta \text{ is ground}\}$. Intuitively speaking, the comparisons in such a subset are considered to be true, and all others are false. A rule with a comparison in its body that is not in this

subset can hence not be used to determine the truth value of atoms. Both the deterministic rules $\mathcal{R}_L$ and the subset $C_M$ together define a set of all ground atoms $h\theta$ that are derivable, i.e., entailed by the program, and thus considered true. Such a set is called a *possible world*. We refer the reader to the paper of De Raedt and Kimmig (2015) for a detailed account of possible worlds in a PLP context. Following the distribution semantics of Sato (1995) and by taking inspiration from Gutmann et al. (2011), we define the probability of such a possible world.

**Definition 3.5** (Probability of a possible world). Let $\mathbb{P}$ be a ground DeepSeaProbLog program and $C_M = \{c_1, \ldots, c_H\} \subseteq \mathcal{C}_M$ a set of PCFs whose elements depend on the random variables declared in the set of distributional facts $\mathcal{F}_D$. The probability of a world $\omega_{C_M}$ is then defined as

$$P(\omega_{C_M}) = \int \left[ \left( \prod_{c_i \in C_M} \mathbb{1}(c_i) \right) \left( \prod_{c_i \in \mathcal{C}_M \setminus C_M} \mathbb{1}(\bar{c}_i) \right) \right] \, \mathrm{d}P_{\mathcal{F}_D}. \tag{3.1}$$

Here the symbol $\mathbb{1}$ denotes the indicator function, $\bar{c}_i$ expresses the complement of the comparison $c_i$ and $\mathrm{d}P_{\mathcal{F}_D}$ represents the joint probability measure of the random variables defined in the set of distributional facts $\mathcal{F}_D$.

**Example 3.2** (Probability of a possible world). Given $\mathbb{P}$ as in Example 3.1, where `temperature_predictor(Data)` predicts the tuple $(\mu(\texttt{Data}), \sigma(\texttt{Data}))$, the probability of the possible world $\omega_{\{\texttt{temp(data1)}>20\}}$ is given by

$$\int \mathbb{1}(x>20) \frac{1}{\sqrt{2\pi}\sigma(\texttt{data1})} \exp\left( -\frac{(x-\mu(\texttt{data1}))^2}{2\sigma^2(\texttt{data1})} \right) \, \mathrm{d}x. \tag{3.2}$$

Indeed, the measure $\mathrm{d}P_{\mathcal{F}_D}$ decomposes into a probability distribution $w(\boldsymbol{x})$ and a differential $\mathrm{d}x$. In Example 3.1, this distribution $w(\boldsymbol{x})$ is exactly the normal distribution, while the product of PCFs in Equation 3.1 reduces to just a single indicator of the PCF $(x > 20)$.

**Definition 3.6** (Probability of query atom). The probability of a ground atom $q$ is given by

$$P(q) = \sum_{C_M \subseteq \mathcal{C}_M : q \in \omega_{c_M}} P(\omega_{C_M}). \tag{3.3}$$

**Proposition 3.1** (Measureability of query atom). Let $\mathbb{P}$ be a valid DeepSeaProbLog program, then $\mathbb{P}$ defines, for an arbitrary query atom $q$, the probability that $q$ is true.

*Proof.* See Appendix B. □

## 4 INFERENCE AND LEARNING

### 4.1 INFERENCE VIA REDUCTION TO WEIGHTED LOGIC

A popular technique to perform inference in probabilistic logic programming uses a reduction to so-called *weighted model counting* (WMC); instead of computing the probability of a program, one computes the weight of a propositional logical formula (Chavira and Darwiche, 2008; Fierens et al., 2015). For DeepSeaProbLog, the equivalent approach is to map a ground program onto a *satisfiability modulo theory* (SMT) formula (Barrett and Tinelli, 2018). The analogous concept to WMC for these formulas is *weighted model integration* (WMI) (Belle et al., 2015; Morettin et al., 2021), which can handle infinite sample spaces. In all that follows, for ease of exposition, we assume that all joint probability distributions are continuous. This can, however, be generalised to discrete distributions by either allowing for Dirac delta distributions or taking a measure theoretic approach (Miosic and Zuidberg Dos Martires, 2021).

**Proposition 4.1** (Inference as WMI). Let us assume that the measure $\mathrm{d}P_{\mathcal{F}_D}$ decomposes into a joint probability density function $w(\boldsymbol{x})$ and a differential $\mathrm{d}\boldsymbol{x}$. The probability of a query atom can then be expressed as the weighted model integration problem

$$P(q) = \int \left[ \sum_{C_M \subseteq \mathcal{C}_M : q \in \omega_{c_M}} \prod_{c_i \in C_M \cup \overline{C}_M} \mathbb{1}(c_i(\boldsymbol{x})) \right] w(\boldsymbol{x}) \, \mathrm{d}\boldsymbol{x}, \tag{4.1}$$

where $\overline{C}_M := \{\bar{c}_i \mid c_i \in \mathcal{C}_M \setminus C_M\}$.

*Proof.* See Appendix C. □

Being able to express the probability of a queried atom in DeepSeaProbLog as a weighted model integral allows us to adapt and deploy inference techniques developed in the weighted model integration literature to perform inference in DeepSeaProbLog. More concretely, the approximate inference algorithm Sampo presented in (Zuidberg Dos Martires et al., 2019) will be used. The idea is to take the sum of products of indicator functions present in Equation 4.1 and rewrite it as recursively nested sums of products. This process is also referred to as knowledge compilation (Darwiche and Marquis, 2002), a state-of-the-art technique for probabilistic inference (Chavira and Darwiche, 2008; Fierens et al., 2015). Furthermore, as the integral, i.e., the expected value of the sums of products of indicator functions, is usually intractable, we approximate it by sampling values from the joint probability distribution as

$$P(q) = \int \text{SP}(\boldsymbol{x}) \cdot w(\boldsymbol{x}) \, \mathrm{d}\boldsymbol{x} \approx \frac{1}{|\mathcal{X}|} \sum_{\boldsymbol{x} \in \mathcal{X}} \text{SP}(\boldsymbol{x}) \qquad (4.2)$$

where $\mathcal{X}$ denotes a set of samples drawn from $w(\boldsymbol{x})$ and $\text{SP}(\boldsymbol{x})$ denotes the sum of products of indicator functions (cf. the sum-product expression inside the brackets of Equation 4.1).

Note that the Sampo algorithm only samples random variables whose expected value with respect to the function $\text{SP}(\boldsymbol{x})$ cannot be computed exactly. Hence, DeepSeaProbLog is able to perform exact symbolic inference for random variables with finite sample spaces, e.g., Boolean random variables. In turn, this means that in the absence of random variables with infinite sample spaces an implementation of DeepSeaProbLog using Sampo coincides with DeepProbLog on a semantics level (Proposition A.1) as well as on an inference level. In Appendix D we provide a diagrammatic representation of the function $\text{SP}(\boldsymbol{x})$ for the query in Example 3.1 where we also perform exact symbolic inference for the discrete variable.

## 4.2 DIFFERENTIATING A WEIGHTED MODEL INTEGRAL

Neural networks in a DeepSeaProbLog program depend on a set of parameters $\boldsymbol{\Lambda} := \cup_i \{\boldsymbol{\lambda}_i\}_{i=1}^K$ (cf. Definition 3.2). In order to perform learning in DeepSeaProbLog, we need to take the gradient of a loss function that compares the probability $P(q)$ to a training signal. More precisely, we need to compute the derivative

$$\partial_\lambda \mathcal{L}(P_{\boldsymbol{\Lambda}}(q)) = \partial_{P_{\boldsymbol{\Lambda}}(q)} \mathcal{L}(P_{\boldsymbol{\Lambda}}(q)) \cdot \partial_\lambda P_{\boldsymbol{\Lambda}}(q), \qquad (4.3)$$

where we now explicitly indicate the dependency of the probability on $\boldsymbol{\Lambda}$ and where $\lambda \in \boldsymbol{\Lambda}$. This means that we need to be able to differentiate the probability of a query of interest $P_{\boldsymbol{\Lambda}}(q)$ with respect to $\lambda$, which presents two obstacles. First, the question of differentiating through the sampling process of Equation 4.2 and second, the non-differentiability of the indicator functions in $\text{SP}(\boldsymbol{x})$.

The non-differentiability of sampling is tackled using the reparametrization trick (Ruiz et al., 2016). Reparametrization offers better estimates than other approaches, such as REINFORCE (Williams, 1992) and is readily utilised in modern probabilistic programming languages such as Tensorflow Probability (Tran et al., 2017) and Pyro (Bingham et al., 2019). In particular, the reparametrization $\boldsymbol{x} = r(\boldsymbol{u}, \boldsymbol{\Lambda})$, with $\boldsymbol{u} \sim p(\boldsymbol{u})$, allows us to write $\partial_\lambda P_{\boldsymbol{\Lambda}}(q)$ as

$$\partial_\lambda P_{\boldsymbol{\Lambda}}(q) = \partial_\lambda \int \text{SP}(\boldsymbol{x}) \cdot w_{\boldsymbol{\Lambda}}(\boldsymbol{x}) \, \mathrm{d}\boldsymbol{u} = \partial_\lambda \int \text{SP}(r(\boldsymbol{u}, \boldsymbol{\Lambda})) \cdot p(\boldsymbol{u}) \, \mathrm{d}\boldsymbol{u}. \qquad (4.4)$$

Conversely, the non-differentiability of the indicator functions prevents us from swapping the order of differentiation and integration, following Leibniz' integral rule (Flanders, 1973). We resolve this technical issue by applying continuous relaxations of the indicator functions following the work of Petersen et al. (2021). These relaxations then yield the approximation

$$\partial_\lambda P_{\boldsymbol{\Lambda}}(q) \approx \partial_\lambda \int \text{SP}_s(r(\boldsymbol{u}, \boldsymbol{\Lambda})) \cdot p(\boldsymbol{u}) \, \mathrm{d}\boldsymbol{u} \approx \int [\partial_\lambda \text{SP}_s(r(\boldsymbol{u}, \boldsymbol{\Lambda}))] \cdot p(\boldsymbol{u}) \, \mathrm{d}\boldsymbol{u}, \qquad (4.5)$$

where the subscript $s$ in $\text{SP}_s(\boldsymbol{x})$ denotes the continuously relaxed or 'softened' version of $\text{SP}(\boldsymbol{x})$. For example, the indicator of a PCF ($g(\boldsymbol{x}) > 0$) is relaxed into the sigmoid $\sigma(\beta \cdot g(\boldsymbol{x}))$. Here. the *coolness* parameter $\beta \in (0, +\infty)$ is the inverse of the temperature of the relaxation and determines the strictness of the relaxation: for $\beta \to +\infty$ we recover the hard indicator function. In Appendix E we provide a more detailed account covering relaxations of further PCFs. Note that the relaxation of the indicator functions results in a biased expression for $\partial_\lambda P_{\boldsymbol{\Lambda}}(q)$ that only vanishes in the infinite coolness limit.

**Proposition 4.2** (Unbiasedness in the infinite coolness limit). Let $\mathbb{P}$ be a DeepSeaProbLog program and $q$ a query atom with PCFs ($g_i(\boldsymbol{x}) \bowtie 0$) and corresponding coolness parameters $\beta_i$. If $\partial_\lambda(g_i \circ r)$ is locally integrable over $\mathbb{R}^k$ and every $\beta_i \to +\infty$, then

$$\partial_\lambda P(q) = \int \partial_\lambda \text{SP}_s(r(\boldsymbol{u}, \boldsymbol{\Lambda})) \cdot p(\boldsymbol{u}) \, \mathrm{d}\boldsymbol{u}. \qquad (4.6)$$

*Proof.* The proof makes use of the mathematical theory of distributions (Schwartz, 1957), which generalise the concept of functions, and is given in Appendix F. □

Petersen et al. (2021) already stated in their work that, in the infinite coolness limit, a relaxed function coincides with the non-relaxed one. Proposition 4.2 extends this result by stating that this property also holds for the derivatives of relaxed and non-relaxed functions.

Finally, we estimate the derivative $\partial_\lambda P_\Lambda(q)$ using a set of samples $\mathcal{U}$ drawn from $p(\boldsymbol{u})$.

$$\partial_\lambda P(q) \approx \int [\partial_\lambda \mathrm{SP}_s(r(\boldsymbol{u}, \boldsymbol{\Lambda}))] \cdot p(\boldsymbol{u}) \, \mathrm{d}\boldsymbol{u} \approx \frac{1}{|\mathcal{U}|} \sum_{\boldsymbol{u} \in \mathcal{U}} \partial_\lambda \mathrm{SP}_s(r(\boldsymbol{u}, \boldsymbol{\Lambda})). \qquad (4.7)$$

Note how Equation 4.7 is just the derivative of the original inference approximation from Equation 4.2, but with reparametrization and continuous relaxations applied.

## 5 RELATED WORK

From a NeSy perspective the formalism most closely related to DeepSeaProbLog is that of *Logic Tensor Networks* (LTNs) (Donadello et al., 2017; Badreddine et al., 2022). The main difference between LTNs and DeepSeaProbLog is the fuzzy logic semantics of the former and the probabilistic semantics of the latter. Interestingly, both systems use similar continuous relaxations when differentiating through comparisons of continuous variables, which is also in line with other NeSy approaches based on fuzzy logics (Marra et al., 2019). However, fuzzy-based approaches require these relaxations at the semantics level, in contrast to DeepSeaProbLog. LTNs' fuzzy semantics do also exhibit drawbacks on a practical level. Unlike DeepSeaProbLog with its probabilistic semantics, LTNs are not capable of performing neural-symbolic generative modelling (cf. Section 6.2) or density estimation (cf. Section 6.3). For a broader overview of the field of neural-symbolic AI, we refer the reader to a series of survey papers that have been published in recent years (Garcez et al., 2019; Marra et al., 2021; Garcez et al., 2022; Giunchiglia et al., 2022).

From a probabilistic programming perspective, DeepSeaProbLog is related to languages that handle discrete and continuous random variables such as *BLOG* (Milch, 2006; Wu et al., 2018), *Distributional Clauses* (Gutmann et al., 2011) and *Anglican* (Tolpin et al., 2016; Staton et al., 2016), which have all been given declarative semantics, i.e., the meaning of the program does not depend on the underlying inference algorithm. Treating discrete variables as first-order citizens comes with the drawback of non-differentiability, which is a desirable property for neural-symbolic programming. In DeepSeaProbLog we circumvent non-differentiability by introducing continuous relaxations, while at the same time retaining declarative semantics. We stress that DeepSeaProbLog's semantics do not only define the meaning of a probabilistic query in a declarative fashion (Equation 3.3) but also the meaning of its gradient (Equation 4.5). This stands in stark contrast to end-to-end (deep) probabilistic programming languages such as Pyro (Bingham et al., 2019) or Tensorflow Probability (Dillon et al., 2017), which have only been equipped with operational semantics.

An interesting direction for future research is the adaption of advanced inference techniques already present in deep probabilistic programming languages and which usually require differentiability, e.g., *stochastic variational inference* (Hoffman et al., 2013) or *NUTS Hamiltonian Monte Carlo* (Hoffman et al., 2014) inference. As our current implementation of DeepSeaProbLog already uses TensorFlow Probability as its arithmetic engine in the back-end, this should be an attainable objective that we leave for future work.

## 6 EXPERIMENTAL EVALUATION

We have two main experimental questions. **(Q1)** Is learning, which includes inference, with continuous relaxations and reparametrizations possible? **(Q2)** Does DeepSeaProbLog bridge the DPP-NeSy gap? We answer **(Q1)** on the newly-introduced MNIST subtraction task (cf. Section 6.1) and a neural hybrid Bayesian net (cf. Section 6.3). **(Q2)** will be answered by introducing *neural-symbolic variational auto-encoders*, inspired by the work of Misino et al. (2022).

The details of our experimental setup, such as the used hardware, the annealing scheme used for the coolness, and hyperparameters used for the neural networks are given in Appendix G.

## 6.1 NEURAL-SYMBOLIC OBJECT DETECTION

It is difficult to compare the performance of DeepSeaProbLog to other, existing methods. Hence, we introduce the MNIST subtraction task, which can be solved by other neural-symbolic systems and purely neural approaches. Given a single image containing two MNIST digits, the task is to predict the correct value of the subtraction of those digits. It is similar to the MNIST addition experiment of Manhaeve et al. (2018), yet we introduce an additional difficulty by requiring segmentation of the given images. The segmentation problem will be solved by combining the localisation and classification power of a simple, two-stage object detector inspired by Ren et al. (2015) with DeepSeaProbLog's continuous reasoning capabilities. Specifically, since the subtraction is non-commutative and so location dependent, continuous and discrete knowledge can be intertwined by connecting spatial reasoning on the given image with the discrete digit predictions (Listing 1). We will see that DeepSeaProbLog provides better detections by exploiting the full support of its predicted bounding boxes for spatial reasoning, in contrast to the usual point estimates of other methods .

```
region(Im, ID, XY) ~ generalisednormal(region_dimensions(Im, ID, XY)).
object(Im, ID) ~ bernoulli(region_score(Im, ID)).
digit(Im, ID) ~ categorical(d_classifier(Im, ID), [0,...,9]).

subtraction(Im, Diff, Dist) :-
    object(Im, ID1), object(Im, ID2), ID1 =\= ID2,
    region(Im, ID1, y) =:= region(Im, ID2, y),
    distance(Im, ID1, ID2, PredDist), PredDist =:= Dist,
    region(Im, ID1, x) < region(Im, ID2, x),
    Diff is digit(Im, ID1) - digit(Im, ID2).
```

Listing 1: The first NDF represents the x or y location of bounding box ID as a generalised normally distributed random variable (Nadarajah, 2005) with mean and scale being the center and width of the box, respectively. object(Im, ID) indicates whether there is an object in box ID while digit(Im, ID) classifies the content of box ID into the possible digit classes. The predicate subtraction(Im, Diff) looks at all combinations of different bounding boxes that contain an object and uses spatial reasoning via multiple PCFs to determine which box corresponds to which digit, after which a prediction for Diff can be given.

Notice that the supervision on the coordinates of the bounding boxes is underspecified, as the distance and left-right relation between two digits is only sufficient to deduce relative positions. No additional learning signal from the classifier can resolve this underspecification due to the discontinuity between the box regression and classification networks of two-stage object detectors (Ren et al., 2015). We solve this issue by borrowing an idea from the continual learning community (De Lange et al., 2021) and include a small set of memory samples with direct supervision on the coordinates of both digits in the images. The memory is small enough such that its optimisation alone is insufficient to solve the overall problem.

Alternatively, DeepSeaProbLog can offer a possible solution to the problem of joint approximate training which has plagued two-stage object detection approaches (Ren et al., 2015) through probabilistic masking. For instance, in Listing 1 we use the generalised normal distribution that allows us to pass a gradient signal through the masking operation. This is not possible with the usual hard masking. Results show that probabilistic masking leads to a significant increase in accuracy and IoU for **(E2)**. These results and more details on the experimental setup can be found in Appendix G.1.

We compare DeepSeaProbLog to a neural baseline and LTNs on two experimental cases. In the first, **(E1)**, we train on a data set containing all 100 possible differences between two digits and giving their subtraction results as supervision. The test and validation data also contain all 100 possible differences. In **(E2)**, we only provide 70 out of the 100 possible differences during training while distributing the remaining 30 among the validation (10) and test data (20).

The most striking observation in our results (Table 1) is the poor performance of the neural baseline, especially when considering experiment **(E2)**. In essence, the neural baseline fails to generalise the learned knowledge. While both NeSy methods are able to generalise, DeepSeaProbLog distinguishes itself by better accuracies. The reason seems also clear; DeepSeaProbLog can exploit the full support of its continuous distributions to reason over the bounding boxes, leading to higher IoU values. Since classification depends on good bounding boxes, the higher IoU can explain the increase in accuracy.

Table 1: Median accuracy and Intersection-over-Union (IoU) for classifying the result of the difference of two digits in one image. The sub- and superscripts indicate 25% and 75% quantiles, respectively, taken over 10 training runs. The quantiles represent the boundaries between which the middle 50% of observed accuracy values lie. Do note that the results of **(E1)** and **(E2)** should not be compared, as they are computed on different test sets. All results are reported in percentages.

| Experiment | DeepSeaProbLog | | LTN | | Neural baseline | |
|---|---|---|---|---|---|---|
| | acc. | IoU | acc. | IoU | acc. | IoU |
| **(E1)** | $88.16^{+0.22}_{-1.88}$ | $55.70^{+3.50}_{-1.82}$ | $72.88^{+1.72}_{-1.12}$ | $52.29^{+0.94}_{-0.39}$ | $72.20^{+2.78}_{-1.44}$ | $52.46^{+1.81}_{-1.61}$ |
| **(E2)** | $87.09^{+1.81}_{-1.09}$ | $56.10^{+2.32}_{-3.00}$ | $76.73^{+1.36}_{-1.18}$ | $52.30^{+1.75}_{-1.44}$ | $0.27^{+0.09}_{-0.07}$ | $54.23^{+2.88}_{-1.43}$ |

## 6.2 NEURAL-SYMBOLIC VARIATIONAL AUTO-ENCODER

In the previous experiment we predicted the difference between two digits by regarding it as a classification problem. Now we would like to run this experiment in reverse: generate two MNIST digits given a subtraction result. Inspired by the work of Misino et al. (2022), we opt for a conditional variational auto-encoder approach (CVAE) (Kingma and Ba, 2015; Sohn et al., 2015). A diagrammatic overview is given in Figure 1.

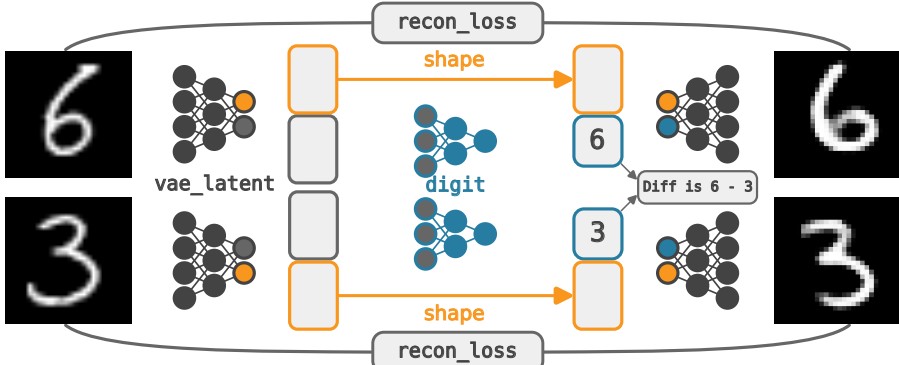

Figure 1: Each image is encoded into a multivariate normal NDF (`shape`) and a latent vector by `vae_latent`. The latter forms the input to the NDF `digit`, while a sample of the former is combined with the outcome of `digit` to form the input of the decoder. Doing so for both images yields two reconstructions, which are compared to the original images in a probabilistic `recon_loss`. Note that the values of `digit` for both images also have to comply to the value of the given difference.

The flexibility of DeepSeaProbLog allows to declaratively encode the architecture presented in Figure 1 as a neural probabilistic logic program. The probabilistic aspect is crucial as other NeSy frameworks, such as LTNs, lack the probabilistic semantics needed to express such deep, relational generative models. Details on our NeSy-VAE implementation can be found in Appendix G.2.

We jointly train the encoder, decoder and digit classifier using the input images themselves as supervision in a reconstruction loss. Additionally, the result of the difference between the digits has to comply with a given value. For instance, if images 6 and 3 are given, then we give the label 3 as additional supervision.

After the training phase, pairs of digits that result in a specific subtraction result can now be generated. To do this, we first sample from the normally distributed latent space of the NeSy-VAE, which generates two latent representations for two digits. Next, the logic deduces which digits comply with the given subtraction result and attaches these to the two latent representations. Finally, these representations are passed through the learned decoder, which constructs two images that satisfy the subtraction result $\boxed{?}-\boxed{?}=5$ (right).

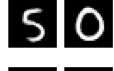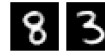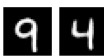

While our NeSy-VAE is inspired by the VAEL architecture of Misino et al. (2022), conceptual differences exist. Most notably, for VAEL, the image generation resides outside of the probabilistic logic program. This is in contrast to DeepSeaProbLog where the VAE latent space is an integral part of the deep relational model.

As such, DeepSeaProbLog easily generalises to conditional generative queries that differ significantly from the original optimisation task. More precisely, without performing any retraining, we query the DeepSeaProbLog program to generate a subtrahend given a minuend (top) and the subtraction result, i.e., fill in the blank in $\boxed{/}-\boxed{?} = \texttt{Diff}$ (bottom row). Additionally, we will demand that the generated image is in the same style of writing as the given image through DeepSeaProbLog's reasoning capabilities. From left to right, generated subtrahend images for a $\texttt{Diff}$ value of 0, -5 and -7 respectively.

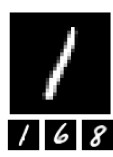

Having demonstrated the generative and reasoning capabilities of the presented NeSy-VAE, we conclude that DeepSeaProbLog bridges the DPP-NeSy gap and we affirmatively answer **(Q2)** as well.

### 6.3 NEURAL HYBRID BAYESIAN NETWORKS

Hybrid Bayesian networks (Lerner, 2003) are probabilistic graphical models that combine discrete and continuous random variables logically. DeepSeaProbLog allows for the introduction of neural components to such models, as shown in Example 3.1. We further extend this example (right) such that the parameters of the variables $\texttt{humid}$ (**H**), $\texttt{cloudy}$ (**C**) and $\texttt{temperature}$ (**T**) are determined by neural networks from sub-symbolic inputs. We opt for the most distant supervision by only giving the probability of $\texttt{enjoy\_weather}$ (**E**) being true or false. The temperature is made inherently probabilistic through the addition of Gaussian noise, which we can model explicitly as a learnable parameter in DeepSeaProbLog.

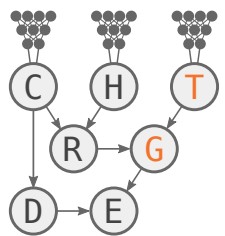

Our optimised neural Bayesian model can be evaluated in to ways. First, the accuracy scores of the networks utilised in $\texttt{cloudy}$ and $\texttt{humid}$, which were $99.46^{+0.13}_{-0.17}$ and $99.20^{+0.40}_{-0.00}$, respectively. Second, the MSE between the true and predicted mean values for $\texttt{temperature}$, being $0.0877^{+0.0268}_{-0.0177}$. Importantly, DeepSeaProbLog was able to get close to the correct amount of noise on $\texttt{temperature}$ from the distant supervision, deviating by $1.10^{+0.14}_{-0.12}$.

Note how this neural and hybrid Bayesian model is a prototypical example of how to exploit the complex dependencies of a probabilistic world. It can be seen as the first step towards being able to successfully apply neural-symbolic principles to domains such as robotics, which involves logical reasoning over discrete and continuous random variables. While DeepSeaProbLog currently lacks the capacity to deal with the dynamics of such an application, it is a promising language in which to symbolically model the uncertain environment of such an autonomous agent.

## 7 CONCLUSION

We presented DeepSeaProbLog, a novel neural-symbolic probabilistic logic programming language that integrates hybrid probabilistic logic and neural networks. Inference is dealt with efficiently through approximate weighted model integration while learning is facilitated by reparametrization and continuous relaxations of non-differentiable logic components. Our experiments illustrated how DeepSeaProbLog is capable of intricate probabilistic modelling allowing for meaningful weak supervision while maintaining strong out-of-distribution performance. Moreover, they showed how hybrid probabilistic logic can be used as a flexible structuring formalism for the neural paradigm that can effectively optimise and reuse neural components in different tasks.

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

## A  SPECIAL CASES OF DEEPSEAPROBLOG

The syntax and semantics of DeepSeaProbLog generalise a number of probabilistic logic programming dialects. For instance, if we assume no dependency of the distributional facts on input data or external neural functions, we obtain a language equivalent to Gutmann et al.'s *Distributional Clauses* (DC) (Gutmann et al., 2011) when restricted to distributional facts. Finally, if we allow for data dependent neural functions in the NDFs but restrict them to Bernoulli and categorical distributions, we obtain Manhaeve et al.'s DeepProbLog (Manhaeve et al., 2018) as a special case.

**Proposition A.1** (DeepSeaProbLog strictly generalises DeepProbLog). DeepProbLog is a strict subset of DeepSeaProbLog where the set of comparison predicates is restricted to $\{=:=\}$, comparisons involve exactly one random variable and the measure $\mathrm{d}P_{\mathcal{F}_D}$ factorizes as a product of independent Bernoulli measures $\prod_{i:\mathrm{x}_i\sim\mathrm{b}_i\in\mathcal{F}_D}\mathrm{d}P_{b_i}$. The subscript on $\mathrm{d}P_{b_i}$ explicitly identifies the measure as the $i^{\text{th}}$ Bernoulli measure and the indices of the product go over all the (Bernoulli) random variables defined in the set of distributional facts $\mathcal{F}_D$.

*Proof.*  We prove Proposition A.1 by showing that applying the restrictions on the constraints and measure in a DeepSeaProbLog program leads to possible worlds that have the same probability of being true as in DeepProbLog. First we write down the definition of the probability of a possible world in a DeepSeaProbLog program.

$$P(\omega_{C_M}) = \int \left[\left(\prod_{c_i\in C_M}\mathbb{1}(c_i)\right)\left(\prod_{c_i\in\mathcal{C}_M\setminus C_M}\mathbb{1}(\bar{c}_i)\right)\right]\mathrm{d}P_{\mathcal{F}_D} \tag{A.1}$$

Now observe that, since there are only Bernoulli distributions, we only need to consider two possible outcomes of a random variable $\mathrm{x}_i$, either zero or one. Therefore, only two kinds of comparisons are present in the program, $\mathrm{x}_i=:=0$ or $\mathrm{x}_i=:=1$ (remember that we restrict ourselves to univariate comparisons). Now note that the following equivalence $\mathrm{x}_i=:=1 \leftrightarrow \neg(\mathrm{x}_i=:=0)$ holds, which means that we can arbitrarily limit comparisons to one of the two possible outcomes of a random variable, e.g., $\mathrm{x}_i=:=0$.

This equivalence can be used to replace the constraints $c_i$ in Equation A.1 by equality constraints involving comparisons to the zero outcome, i.e.,

$$P(\omega_{C_M}) = \int \left[\left(\prod_{i:c_i\in C_M}\mathbb{1}(x_i{=}0)\right)\left(\prod_{i:c_i\in\mathcal{C}_M\setminus C_M}\mathbb{1}(x_i{\neq}0)\right)\right]\prod_{i:\mathrm{x}_i\sim\mathrm{b}_i\in\mathcal{F}_D}\mathrm{d}P_{b_i}, \tag{A.2}$$

where the factorisation of the measure was also applied. Next, we introduce the following notation for the random variables present in the set of constraints $C_M$ and $\mathcal{C}_M\setminus C_M$:

$$\boldsymbol{x}^+ := x_i : c_i \in C_M \tag{A.3}$$
$$\boldsymbol{x}^- := x_i : c_i \in \mathcal{C}_M\setminus C_M \tag{A.4}$$

Note that we only need to consider the case where $\boldsymbol{x}^+\cap\boldsymbol{x}^-=\emptyset$, as otherwise the probability of the possible world would simply be zero and would not contribute to the overall probability of the query atom. Because of this, we can further factorize the measure as

$$\prod_{i:\mathrm{x}_i\sim\mathrm{b}_i\in\mathcal{F}_D}\mathrm{d}P_{b_i} = \underbrace{\left(\prod_{i:c_i\in C_M}\mathrm{d}P_{b_i}\right)}_{=:\mathrm{d}P^+}\underbrace{\left(\prod_{i:c_i\in\mathcal{C}_M\setminus C_M}\mathrm{d}P_{b_i}\right)}_{=:\mathrm{d}P^-}, \tag{A.5}$$

so the integral of a product in Equation A.2 can be rewritten as the product of integrals

$$P(\omega_{C_M}) \tag{A.6}$$
$$= \left[\int\left(\prod_{i:c_i\in C_M}\mathbb{1}(x_i{=}0)\,\mathrm{d}P^+\right)\right]\left[\int\left(\prod_{i:c_i\in\mathcal{C}_M\setminus C_M}\mathbb{1}(x_i{\neq}0)\,\mathrm{d}P^-\right)\right],$$

We have two integrals with integrands that are a product of univariate comparisons. In other words, the factors are all independent. Furthermore, we have a Bernoulli product measure, which means that we can again push the integral inside the product to yield

$$P(\omega_{C_M}) = \tag{A.7}$$

$$\left[ \prod_{i:c_i \in C_M} \left( \int \mathbb{1}(x_i=0) \, \mathrm{d}P^+ \right) \right] \left[ \prod_{i:c_i \in \mathcal{C}_M \backslash C_M} \left( \int \mathbb{1}(x_i \neq 0) \, \mathrm{d}P^- \right) \right]. \tag{A.8}$$

At this point we can simply perform the integrations and obtain

$$P(\omega_{C_M}) = \prod_{i:c_i \in C_M} p_i \prod_{i:c_i \in \mathcal{C}_M \backslash C_M} (1 - p_i), \tag{A.9}$$

which coincides with the probability of a possible world in DeepProbLog (cf. (Manhaeve et al., 2021a, Section 3)). □

Proposition A.1 can easily be extended to also allow for measures of finite categorical distributions, which then translates to (neural) annotated disjunctions. Consequently, as DeepProbLog is a strict superset of ProbLog (Fierens et al., 2015), DeepSeaProbLog also strictly generalises ProbLog.

## B  PROOF OF PROPOSITION 3.1

**Proposition 3.1** (Measureability of query atom). Let $\mathbb{P}$ be a valid DeepSeaProbLog program, then $\mathbb{P}$ defines, for an arbitrary query atom $q$, the probability that $q$ is true.

*Proof.*  DeepSeaProbLog is in essence a subset of the probabilistic logic programming language defined by Gutmann et al. (2011) – the only difference being that the parameters on the right-hand side of a neural distributional fact are not limited to numerical constants any more but can be arbitrary numeric terms. Under the condition that all NDFs and PCFs are valid, this does, however, not violate any of the assumptions made in (Gutmann et al., 2011, Proposition 1) (proving the measurability of a program). We can, hence, conclude that a valid DeepSeaProbLog program induces a probability measure for $q$. □

Note that, similar to ProbLog and DeepProbLog, the semantics for DeepSeaProbLog are only defined for so-called sound programs (Riguzzi and Swift, 2013), which means that all programs become ground eventually when queried.

## C  PROOF OF PROPOSITION 4.1

**Proposition 4.1** (Inference as WMI). Let us assume that the measure $\mathrm{d}P_{\mathcal{F}_D}$ decomposes into a joint probability density function $w(\boldsymbol{x})$ and a differential $\mathrm{d}\boldsymbol{x}$. The probability of a query atom can then be expressed as the weighted model integration problem

$$P(q) = \int \left[ \sum_{C_M \subseteq \mathcal{C}_M : q \in \omega_{\mathcal{C}_M}} \prod_{c_i \in C_M \cup \overline{C}_M} \mathbb{1}(c_i(\boldsymbol{x})) \right] w(\boldsymbol{x}) \, \mathrm{d}\boldsymbol{x}, \tag{4.1}$$

where $\overline{C}_M := \{\bar{c}_i \mid c_i \in \mathcal{C}_M \backslash C_M\}$.

*Proof.*  First, let us consider the indices of the two product expressions in Equation 3.1. We define

$$\overline{C}_M := \{\bar{c}_i \mid c_i \in \mathcal{C}_M \backslash C_M\}$$

such that Equation 3.1 can be rewritten as

$$P(\omega_{C_M}) = \int \left( \prod_{c_i \in C_M \cup \overline{C}_M} \mathbb{1}(c_i(\boldsymbol{x})) \right) \mathrm{d}P_{\mathcal{F}_D} \tag{C.1}$$

Furthermore, decomposing the measure into a probability distribution $w(\boldsymbol{x})$ and the differential $\mathrm{d}\boldsymbol{x}$ of the integration variables yields

$$\int \left( \prod_{c_i \in C_M \cup \overline{C}_M} \mathbb{1}(c_i(\boldsymbol{x})) \right) \cdot w(\boldsymbol{x}) \, \mathrm{d}\boldsymbol{x}. \tag{C.2}$$

We can now plug this last expression into Equation 3.3 resulting in

$$P(q) = \int \sum_{\substack{C_M \subseteq \mathcal{C}_M: \\ q \in \omega_{\mathcal{C}_M}}} \left( \prod_{c_i \in C_M \cup \overline{C}_M} \mathbb{1}(c_i(\boldsymbol{x})) \right) \cdot w(\boldsymbol{x}) \, \mathrm{d}\boldsymbol{x}. \tag{C.3}$$

Note that we changed the order of the integration and summation. This operation was shown to be valid in Zuidberg Dos Martires et al. (2019) using de Finetti's theorem. Zuidberg Dos Martires et al. (2019) also showed that the expression in Equation C.3 is indeed a weighted model integral as defined by Belle et al. (2015). Specifically, line P2 in the proof of Theorem 2 in Zuidberg Dos Martires et al. (2019) corresponds to C.3, which is shown to be equal to an instance of WMI. □

## D SYMBOLIC INFERENCE AND DISCRETE VARIABLES

The inference algorithm of DeepSeaProbLog converts queried probabilistic logic programs to arithmetic circuits (Darwiche and Marquis, 2002). This mechanism is similar to the one present in the implementations of ProbLog2 (Fierens et al., 2015) and DeepProbLog. The circuit then represent the function $\mathrm{SP}(\boldsymbol{x})$ of recursively nested sums of products. We exemplify this conversion on the example program in Listing 2.

```
humid(Data) ~ bernoulli(humidity_detector(Data)).
temp(Data) ~ normal(temperature_predictor(Data)).
snowy_weather ~ beta(2, 7). sunny_weather ~ beta(5, 3).

good_weather(Data, Degree) :-
    humid(Data) =:= 1, temp(Data) < 0, snowy_weather < Degree.
good_weather(Data, Degree) :-
    humid(Data) =:= 0, temp(Data) > 15, sunny_weather > Degree.

query(good_weather(data1, degree1)).
```

Listing 2: Our running `good_weather` example, repeated.

Figure D.2 shows the computation graph obtained from converting the queried program above into an arithmetic circuit. The top '+' node corresponds to the two succeeding branches of `good_weather`. Each of these branches depends on a conjunction of three conjuncts, leading to the three '×' nodes. Each branch terminates at three leaves, two of which contain PCFs with continuous random variables that are being approximated by reparametrised sampling (in orange). The other PCF is replaced by its expected probability obtained via exact symbolic inference, which is again differentiable. Below the leaves, we can see how `data1` forms the input to the neural networks that predict the distributional parameters of `temp(data1)` and the probabilities of `humid(data1)`.

## E DETAILS ON DERIVATIVE ESTIMATE

To give further details on estimating the derivative we will write the expression $\partial_\lambda P_{\boldsymbol{\Lambda}}(q)$ in terms of indicator functions

$$\partial_\lambda P_{\boldsymbol{\Lambda}}(q) = \partial_\lambda \int \mathrm{SP}(\boldsymbol{x}) \cdot w_{\boldsymbol{\Lambda}}(\boldsymbol{x}) \, \partial\boldsymbol{x} \tag{E.1}$$

$$= \partial_\lambda \int \sum_{\substack{C_M \subseteq \mathcal{C}_M: \\ q \in \omega_{\mathcal{C}_M}}} \left( \prod_{c_i \in C_M \cup \overline{C}_M} \mathbb{1}(c_i(\boldsymbol{x})) \right) \cdot w_{\boldsymbol{\Lambda}}(\boldsymbol{x}) \, \mathrm{d}\boldsymbol{x}, \tag{E.2}$$

where the dependency of the probability on the neural parameters $\boldsymbol{\Lambda}$ is again made explicit. Reparametrizing the distribution $w_{\boldsymbol{\Lambda}}(\boldsymbol{x})$ yields

$$\partial_\lambda P_{\boldsymbol{\Lambda}}(q) = \partial_\lambda \int \sum_{\substack{C_M \subseteq \mathcal{C}_M: \\ q \in \omega_{\mathcal{C}_M}}} \left( \prod_{c_i \in C_M \cup \overline{C}_M} \mathbb{1}(c_i(r(\boldsymbol{u}, \boldsymbol{\Lambda}))) \right) \cdot p(\boldsymbol{u}) \, \mathrm{d}\boldsymbol{u}. \tag{E.3}$$

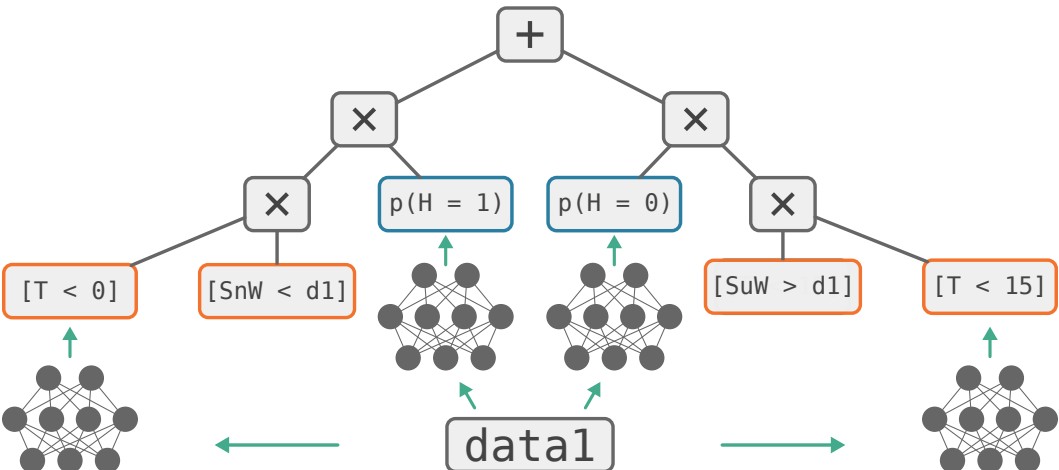

Figure D.2: The result of applying our symbolic inference to the query in Listing 2. The blue boxes are discrete variables, while the orange ones are PCFs with continuous variables. Note that we have abbreviated `temp(data1)` as `T`, `data1` as `d1`, `humid(data1)` as `H`, `snowy_weather` as `SnW` and `sunny_weather` as `SuW`.

Explicitly writing out the indicators clearly illustrates the non-differentiability of $SP(\boldsymbol{x})$, which prevents us from applying Leibniz' integral rule (Flanders, 1973) to swap the order of integration and differentiation. To obtain the necessary differentiability of the integrand, the continuous relaxations introduced by Petersen et al. (2021) are utilised. These relaxations allow for comparison formulae of the form

$$(g(\boldsymbol{x}) \bowtie 0), \quad \text{with} \bowtie \in \{<, \leq, >, \geq, =, \neq\} \tag{E.4}$$

to be relaxed. We write the continuous relaxation of an indicator function $\mathbb{1}(c_i(\boldsymbol{x})) = \mathbb{1}(g_i(\boldsymbol{x}) \bowtie 0)$ as $s_i(\boldsymbol{x})$. Four specific cases of relaxations arise, depending on the comparison operator used. Specifically, we define

$$s_i(\boldsymbol{x}) = \begin{cases} \sigma(\beta_i \cdot g_i(\boldsymbol{x})) & \text{if} \bowtie \in \{>, \geq\}, \\ \sigma(-\beta_i \cdot g_i(\boldsymbol{x})) & \text{if} \bowtie \in \{<, \leq\}, \\ \sigma(\beta_i \cdot g_i(\boldsymbol{x})) \cdot \sigma(-\beta_i' \cdot g_i(\boldsymbol{x})) & \text{if} \bowtie \in \{=\}, \\ 1 - \sigma(\beta_i \cdot g_i(\boldsymbol{x})) \cdot \sigma(-\beta_i' \cdot g_i(\boldsymbol{x})) & \text{if} \bowtie \in \{\neq\}, \end{cases} \tag{E.5}$$

where $\beta_i$ and $\beta_i'$ are the coolness parameters of the continuous relaxations and $\sigma$ denotes the sigmoid function. Note that all four cases originate from the root choice of approximating the step function as a sigmoid function. Additionally, this choice is sound as we have that

$$\lim_{\beta_i \to +\infty} \sigma(\beta_i \cdot g_i(\boldsymbol{x})) = \mathbb{1}(g_i(\boldsymbol{x}) \geq 0). \tag{E.6}$$

Continuously relaxing indicator functions using the definition of Equation E.5 renders the integrand differentiable, allowing the application of Leibniz' integral rule and yielding

$$\partial_\lambda P_{\boldsymbol{\Lambda}}(q) \approx \int \partial_\lambda \sum_{\substack{C_M \subseteq \mathcal{C}_M: \\ q \in \omega_{\mathcal{C}_M}}} \left( \prod_{i: c_i \in C_M \cup \overline{C}_M} s_i(r(\boldsymbol{u}, \boldsymbol{\Lambda})) \right) \cdot p(\boldsymbol{u}) \, \mathrm{d}\boldsymbol{u}. \tag{E.7}$$

The derivative $\partial_\lambda P_{\boldsymbol{\Lambda}}(q)$ can now be computed using off-the-shelf automatic differentiation software such as PyTorch (Paszke et al., 2019) or TensorFlow (Abadi, 2016), which entails that estimating the gradient $\nabla_{\boldsymbol{\Lambda}} P(q) = (\partial_\lambda P(q))_{\lambda \in \boldsymbol{\Lambda}}$ is computationally as expensive as computing the probability itself, up to a constant factor (Griewank and Walther, 2008).

## F   PROOF OF PROPOSITION 4.2

**Proposition 4.2** (Unbiasedness in the infinite coolness limit). Let $\mathbb{P}$ be a DeepSeaProbLog program and $q$ a query atom with PCFs $(g_i(\boldsymbol{x}) \bowtie 0)$ and corresponding coolness parameters $\beta_i$. If $\partial_\lambda(g_i \circ r)$ is locally integrable over $\mathbb{R}^k$ and every $\beta_i \to +\infty$, then

$$\partial_\lambda P(q) = \int \partial_\lambda \mathrm{SP}_s(r(\boldsymbol{u}, \boldsymbol{\Lambda})) \cdot p(\boldsymbol{u}) \, \mathrm{d}\boldsymbol{u}. \tag{4.6}$$

*Proof.*   First we express $P(q)$ using Equation C.3, which we then rewrite without loss of generalisation using only Heaviside distributions[2].

$$P(q) = \int \sum_{\substack{C_M \subseteq \mathcal{C}_M: \\ q \in \omega_{\mathcal{C}_M}}} \left( \prod_{c_i \in C_M \cup \overline{C}_M} \mathbb{1}(c_i(\boldsymbol{x})) \right) \cdot w(\boldsymbol{x}) \, \mathrm{d}\boldsymbol{x} \tag{F.1}$$

$$= \int \sum_{\substack{C_M \subseteq \mathcal{C}_M: \\ q \in \omega_{\mathcal{C}_M}}} \left( \prod_{g_i \in \Sigma_{C_M \cup \overline{C}_M}} H(g_i(r(\boldsymbol{u}, \boldsymbol{\Lambda}))) \right) \cdot p(\boldsymbol{u}) \, \mathrm{d}\boldsymbol{u}. \tag{F.2}$$

In the Equation above, $H(x)$ denotes the Heaviside distribution and $\Sigma_{C_M \cup \overline{C}_M}$ denotes the set of all sigmoid functions involved in the continuous relaxations of the set $C_M \cup \overline{C}_M$.

This rewrite is possible as the indicator function of any PCF $c(\boldsymbol{x})$ is either a step function or decomposes into a product of step functions. Indeed, if the $c(\boldsymbol{x})$ is of the form $g(\boldsymbol{x}) \geq 0$, then $\mathbb{1}(c(\boldsymbol{x})) = H(g(\boldsymbol{x}))$. If it is of the form $g(\boldsymbol{x}) = 0$, then $\mathbb{1}(c(\boldsymbol{x})) = H(g(\boldsymbol{x})) \cdot H(-g(\boldsymbol{x}))$. The other cases with different comparison operators follow from these two.

Differentiating in a distributional sense and applying Leibniz' integral rule (Flanders, 1973) then yields

$$\sum_{\substack{C_M \subseteq \mathcal{C}_M: \\ q \in \omega_{\mathcal{C}_M}}} \left( \sum_{g_j \in \Sigma_{C_M \cup \overline{C}_M}} \int \partial_\lambda H(g_j(r(\boldsymbol{u}, \boldsymbol{\Lambda}))) \cdot \prod_{i \neq j} H(g_i(r(\boldsymbol{u}, \boldsymbol{\Lambda}))) \cdot p(\boldsymbol{u}) \, \mathrm{d}\boldsymbol{u} \right). \tag{F.3}$$

We can reduce the discussion by considering each term in this equation separately, because of the linearity of the integral. In other words, to prove our statement, it suffices to show that

$$\int \partial_\lambda H(g_j(r(\boldsymbol{u}, \boldsymbol{\Lambda}))) \cdot \prod_{i \neq j} H(g_i(r(\boldsymbol{u}, \boldsymbol{\Lambda}))) \cdot p(\boldsymbol{u}) \, \mathrm{d}\boldsymbol{u} \tag{F.4}$$

is equal to

$$\lim_{\beta_1, \ldots, \beta_n \to +\infty} \int \partial_\lambda \sigma(\beta_j \cdot g_j(r(\boldsymbol{u}, \boldsymbol{\Lambda}))) \cdot \prod_{i \neq j} \sigma(\beta_i \cdot g_i(r(\boldsymbol{u}, \boldsymbol{\Lambda}))) \cdot p(\boldsymbol{u}) \, \mathrm{d}\boldsymbol{u}. \tag{F.5}$$

For brevity's sake, we will write the products

$$\prod_{i \neq j} H(g_i(r(\boldsymbol{u}, \boldsymbol{\Lambda}))) \qquad \text{and} \qquad \prod_{i \neq j} \sigma(g_i(r(\boldsymbol{u}, \boldsymbol{\Lambda}))), \tag{F.6}$$

as $\pi_j(\boldsymbol{u})$ and $\pi_j^\sigma(\boldsymbol{u})$, respectively.

Next, using distributional notation, Equation F.4 can be further simplified as

$$\langle \partial_\lambda(H \circ g_j \circ r), \pi_j \cdot p \rangle = \langle \delta \circ g_j \circ r, \partial_\lambda(g \circ r) \cdot \pi_j \cdot p \rangle. \tag{F.7}$$

Note that this expression utilises the assumption that $\partial_\lambda(g_j \circ r) \in L^1_{loc}(\mathbb{R}^k)$, i.e., $\partial_\lambda(g_j \circ r)$ is locally integrable over $\mathbb{R}^k$. This is not a strong assumption, since distributions (generalised functions) are

---

[2]Here we use the term *distribution* in the sense of a generalised function (Schwartz, 1957) and not in the sense of a probability distribution.

only well-defined when acting on functions that are at least locally integrable. Equation F.5 can similarly be rewritten and simplified to obtain the equality

$$\lim_{\beta_1,\ldots,\beta_n\to+\infty} \left\langle \partial_\lambda(\sigma\circ g_j\circ r),\ \pi_j^\sigma\cdot p\right\rangle \tag{F.8}$$

$$= \lim_{\beta_1,\ldots,\beta_{j-1},\beta_{j+1},\ldots,\beta_n\to+\infty} \left\langle \delta\circ g_j\circ r,\ \partial_\lambda(g\circ r)\cdot\pi_j^\sigma\cdot p\right\rangle. \tag{F.9}$$

More explicitly,

$$(\text{F.5}) = \lim_{\beta_1,\ldots,\beta_n\to+\infty} \int \partial_\lambda\sigma(\beta_j\cdot g_j(r(\boldsymbol{u},\boldsymbol{\Lambda})))\cdot\pi_j^\sigma(\boldsymbol{u})\cdot p(\boldsymbol{u})\,\mathrm{d}\boldsymbol{u} \tag{F.10}$$

$$= \lim_{\beta_1,\ldots,\beta_n\to+\infty} \int \frac{l\cdot e^{-g(r(\boldsymbol{u},\boldsymbol{\Lambda}))\cdot\beta_j}}{(1+e^{-g(r(\boldsymbol{u},\boldsymbol{\Lambda}))\cdot\beta_j})^2}\partial_\lambda g_j(r(\boldsymbol{u},\boldsymbol{\Lambda}))\cdot\pi_j^\sigma(\boldsymbol{u})\cdot p(\boldsymbol{u})\,\mathrm{d}\boldsymbol{u} \tag{F.11}$$

$$= \lim_{\beta_1,\ldots,\beta_{j-1},\beta_{j+1},\ldots,\beta_n\to+\infty} \int \delta(g_j(r(\boldsymbol{u},\boldsymbol{\Lambda})))\cdot\partial_\lambda g_j(r(\boldsymbol{u},\boldsymbol{\Lambda}))\cdot\pi_j^\sigma(\boldsymbol{u})\cdot p(\boldsymbol{u})\,\mathrm{d}\boldsymbol{u}. \tag{F.12}$$

The transition from Equation F.11 to Equation F.12 uses the fact that

$$\lim_{\beta_j\to+\infty} \frac{\beta_j\cdot e^{-g(r(\boldsymbol{u},\boldsymbol{\Lambda}))\cdot\beta_j}}{(1+e^{-g(r(\boldsymbol{u},\boldsymbol{\Lambda}))\cdot\beta_j})^2} = \delta(g(r(\boldsymbol{u},\boldsymbol{\Lambda}))), \tag{F.13}$$

in the distributional sense. In addition, we also have (again in the distributional sense) that

$$\lim_{\beta_i\to+\infty} \sigma(\beta_i\cdot g_i(r(\boldsymbol{u},\boldsymbol{\Lambda}))) = H(g_i(r(\boldsymbol{u},\boldsymbol{\Lambda}))). \tag{F.14}$$

This final equation allows us to simplify $\pi_j^\sigma(\boldsymbol{u})$ in Equation F.11 to $\pi_j(\boldsymbol{u})$ by repeating the above steps for each index $i$ separately. Hence, we can conclude that our relaxation of $\partial_\lambda P(q)$ is indeed unbiased in the infinite coolness limit. $\qquad\square$

## G EXPERIMENTAL DETAILS

This section will introduce the DeepSeaProbLog programs, neural network architectures and elaborated figures for each of the experiments present in the main body of the paper. All experiments were run on an HP ZBook Power G8 (NVIDIA T1200 GPU, Intel i9-11900H @ 2.50GHz, 16 GB RAM), except the LTN comparison in Section 6.1. Note that the optimisation of any hyperparameters, such as learning rate, dropout rate or number of training epochs, was done via a grid search on a separate validation set for 10 independent runs.

### G.1 NESY OBJECT DETECTION

**Setup details and DeepSeaProbLog program.** We write the full DeepSeaProbLog program used for the subtraction experiment below. The query `subtraction` is called and optimized for 15 000 training samples of images containing two MNIST digits. Moreover, a set of 100 memory samples with direct supervision on the locations of the bounding boxes of the two digits is used to optimise `location_supervision` in parallel to calibrate the underspecified location predictions. The same set of 100 samples is initially used in a separate curriculum learning phase with additional supervision on the class labels, in order to avoid degenerate solutions. As for validation and test sets, the number of samples in each depends on which concrete setting is considered. For the in-distribution setting, validation and test sets consist of 1000 and 5000 samples, respectively. Conversely, in the out-of-distribution case, the validation set has 1000 samples while the test set has 2000 available.

```
width ~ normal(0, boxwidth).
region(Im, ID, XY) ~ generalisednormal(region_dimensions(Im, ID, XY)).
object(Im, ID) ~ bernoulli(region_score(Im, ID)).
digit(Im, ID) ~ categorical(d_classifier(Im, ID), [0,...,9]).

subtraction(Im, Diff, Dist) :-
```

```
    object(Im, ID1), object(Im, ID2), ID1 =\= ID2,
    region(Im, ID1, y) =:= region(Im, ID2, y),
    distance(Im, ID1, ID2, PredDist), PredDist =:= Dist,
    region(Im, ID1, x) < region(Im, ID2, x),
    Diff is digit(Im, ID1) - digit(Im, ID2).

location_supervision(Im, X1, X2, Y1, Y2) :-
    object(Im, ID1), object(Im, ID2), ID1 =\= ID2,
    region(Im, ID1, x) < region(Im, ID2, x),
    region(Im, ID1, x) =:= X1 + width, region(Im, ID2, x) =:= X2 + width,
    region(Im, ID1, y) =:= Y1 + width, region(Im, ID2, y) =:= Y2 + width.

curriculum(Im, N1, N2, X1, X2, Y1, Y2 ) :-
    object(Im, ID1), object(Im, ID2), ID1 =\= ID2,
    region(Im, ID1, x) < region(Im, ID2, x),
    digit(Im, ID1) =:= N1, digit(Im, ID2) =:= N2,
    region(Im, ID1, x) =:= X1 + width, region(Im, ID2, x) =:= X2 + width,
    region(Im, ID1, y) =:= Y1 + width, region(Im, ID2, y) =:= Y2 + width.
```

**Parameters and neural architectures.** A schematic overview of the neural architecture used for all different methods can be seen in Figure G.3. In the case of the neural-symbolic methods, the output of this architecture is immediately used in further logic. In the case of the neural baseline, the logic part is replaced by a 4-layer fully connected network with 128, 96, 64 and 19 hidden units followed by a softmax activation function. For LTN, the only architectural difference is the output of the location network, which only provides a single point estimate for the position. Finally, the Adam optimiser (Kingma and Ba, 2015) was utilised with a learning rate of $10^{-3}$. DeepSeaProbLog was run for 4 epochs, LTNs were run for 10 epochs while the neural baseline was given 30 epochs, all with a batch size of 10. All methods optimised the cross-entropy loss function, in this and all following experiments. Finally, an annealing scheme for the coolness parameters of our comparison formulae was also used. Specifically, we used a hyperbolic tangent function to scale the coolness parameter between 1 and 8 for the equality comparisons. The inequality comparisons ranged from 1 to 24. Both these values were determined through a grid search on the separate validation set. We performed an annealing step after every batch for 1000 batches after which the maximum coolness (8 or 24) was reached. Once the maximum coolness was reached it stayed constant. Recall that these coolness parameters determine the strictness of the comparisons and, consequently, also control the learning signal that can flow through the comparisons.

**Complications.** The weak supervision on just the result of the difference of two digits can lead to a number of logically equivalent, yet degenerate solutions that do not correspond to our human intuition. For example, if the neural digit classifiers would reverse the order of their classification, e.g., classifying a true 8 as a 1 or a true 5 as a 4, then a correct subtraction result is still given because of the symmetry of the subtraction. Analogously, only knowing the distance between the bounding boxes allows for an infinite amount of correct relative coordinate predictions on the continuous side of the logic. However, these boxes will not correspond to the locations of the digits and are also undesirable solutions. In other words, the distant supervision leaves both the discrete and continuous components underspecified. Such degenerative solutions not only do not align with our human interpretation, but they might also induce a degree of volatility in the learning process. These issues are the reason why a phase of curriculum learning and multi-predicate optimisation as described in our setup was included, as it forces the neural networks into a direction that is consistent with our usual interpretations of the digits and their absolute locations. Similar behaviour was observed by Manhaeve et al. (2021b) where a brief pre-training phase gave a sufficient direction for learning while using their approximate probabilistic inference techniques. The pre-training phase was given 50 epochs using a batch size of 2 for DeepSeaProbLog and the neural baseline. LTNs trained for 100 epochs because of a slower rate of convergence.

**Additional results and interpretations.** Figures G.4 and G.5 show a more detailed evolution of the training process of the different methods. First, they illustrate the flexibility of NeSy methods since the pre-trained networks seemed to have little effect on the neural baseline. In particular, these networks only seem to affect the initial learning stages of the neural baseline, as can be seen from a couple of early peaks in validation accuracy. However, because of the lack of a proper function for

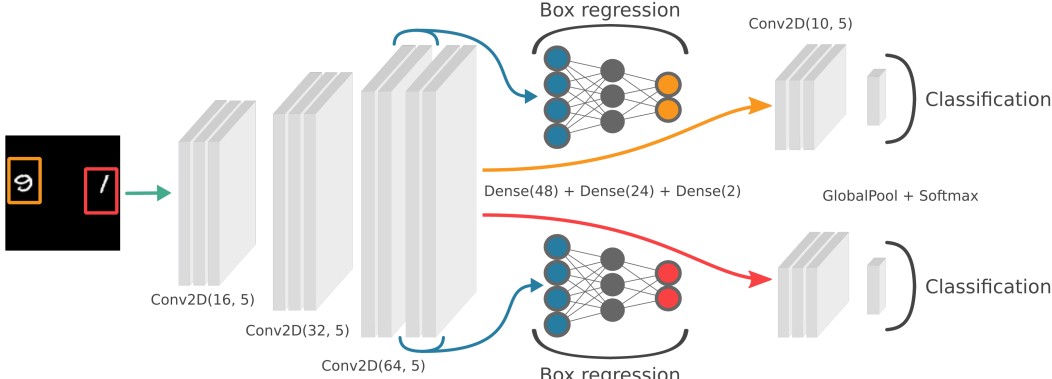

Figure G.3: Overall neural architecture for the subtraction experiment. The initial convolutions are shared by both the classification and regression networks. To get two box predictions, we apply the same 3-layer regression network on each half of the final convolutional output by flattening that output. This can be easily generalised to more boxes for more objects. Given the box predictions, the initial image is masked with these boxes. Both masked images then go to the same convolutional layers again and their full outputs are then put through the classification network to obtain the class predications of every box. All activations functions are set to the ReLU function, except the output of the classification network, which is a softmax function.

the networks, a purely neural optimisation can not fully exploit these pre-trained states. Second, the evolution itself seems to be more consistent for both NeSy methods due to lower variability. Note however that, while LTNs can solve this tasks, they only provide a point estimate without any further indication to the uncertainty on this estimate. DeepSeaProbLog on the other hand models the location with inherent uncertainty. It also has to be mentioned that DeepSeaProbLog and LTNs are still a lot quicker than the neural baseline when looking at actual computation time. Roughly speaking, every 100 iterations took about 20 seconds for DeepSeaProbLog while the neural baseline took around 10 seconds. Given the rate of convergence of both methods (Figure G.4), DeepSeaProbLog is still significantly faster than the neural baseline even though it includes probabilistic logic in its architecture.

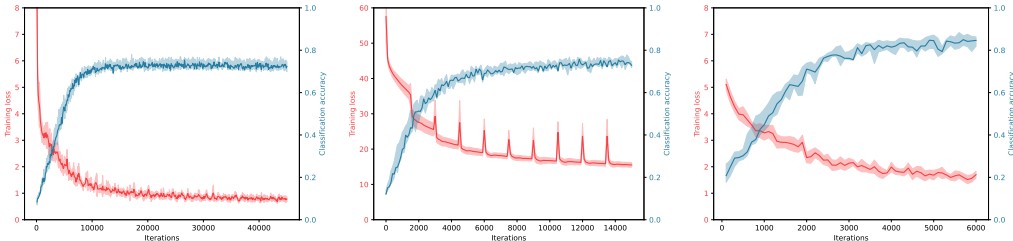

Figure G.4: Neural baseline (left), LTNs (middle) and DeepSeaProbLog (right) training evolution for the in-distribution setting. Note that the neural baseline still requires far more optimisation updates to reach acceptable performance, while still not reaching the same level as DeepSeaProbLog or LTNs. Note that the LTN loss is different in scale as its objective is satisfaction maximization (maxSAT) (Li and Manya, 2021).

**Joint approximate training for two-stage object detectors.** As mentioned in Section 6.1, DeepSeaProbLog can mitigate the discontinuity between the box regression and classification components of a two-stage object detector through probabilistic masking. Specifically, we consider the indices of every pixel as their $x$ and $y$ coordinates. Our predicted generalised normal distributions expresses a two-dimensional probability distribution on the image, hence we can evaluate every pixel according to this distribution. By normalising these evaluations such that the maximum value is 1, we can use them as masking values. In other words, we can apply our two-dimensional generalised

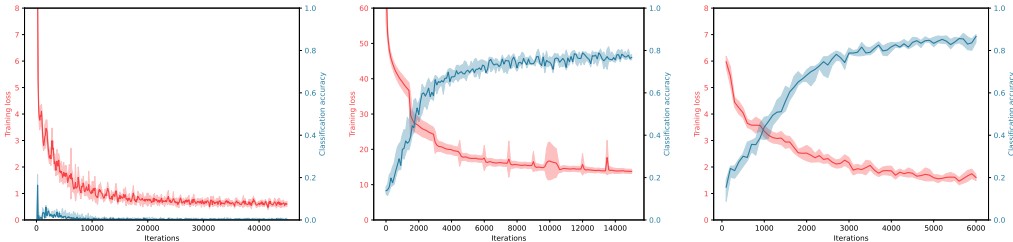

Figure G.5: Training evolution for the neural baseline (left) compard to LTNs (middle) and DeepSeaProbLog (right) in the out-of-distribution setting. The neural baseline fails outright to generalise to the validation and test set.

normal distribution representing our bounding box directly as a mask on the image. Importantly, this procedure is completely differentiable as we evaluate the pixel coordinates according to the parametrized distribution. Moreover, it implies that the continuous reasoning that DeepSeaProbLog does on the distributions corresponds soundly to spatial reasoning on the boxes. Probabilistic masking avoids having to use a specific memory set of samples with direct supervision on the absolute coordinates of the desired bounding boxes, since if the predicted coordinates do not align with the actual digits, the learning signal of the classifier will correct that. Of course, to prevent that the regression predicts a distribution that covers the whole image, we have to regularise the scale of the distributions. This is done by including a PCF that expresses that a generalised normal centered at zero with the predicted scale of the boxes is equal to a generalised normal with a scale value equal to half the width of the desired box. It yields a noticeable improvement to both accuracy and IoU. In the out-of-distribution setting, classification accuracy of the difference is $93.00^{+0.91}_{-0.37}$ while the IoU is $62.46^{3.53}_{-0.82}$.

### G.2 NEURAL-SYMBOLIC VARIATIONAL AUTOENCODER

**Setup details and DeepSeaProbLog programs.** Each data sample consists of 2 regular MNIST digits and the result of their subtraction. The first digit takes the place of the minuend while the second one is interpreted as the subtrahend. The training, validation and test sets had 30 000, 1 000 and 1 000 samples of this form, respectively. Encoding a VAE without additional logic in DeepSeaProbLog is straightforward (Listing 3), while adding logic involves more engineering freedom (Listing 4). We opted for the simplest use of a conditional variational auto-encoder by only using the classified digit as additional input to the decoder. Note that during optimisation, both the VAE and digit classifier are trained jointly.

```
prior ~ normal(0, 1).
vae_latent(Image) ~ normal(vae_encoder(Image)).

good_image(Image) :-
    (vae_latent(Image) =:= prior),
    reconstruction_loss(vae_decoder(vae_latent(Image)), Image).
```

Listing 3: Prototypical implementation of a VAE in DeepSeaProbLog. The encoder and decoder have distinct identifiers, `vae_encoder` and `vae_decoder`. The outputs of the encoder are the means and variances of the latent distributions of the VAE, which are modelled explicitly in DeepSeaProbLog as a series of normally distributed continuous random variables. The definition of a good image is given as an image with a standard normal latent representation and whose decoding unifies with itself.

```
prior1(ID) ~ normal(0, 1). prior2(ID) ~ normal(0, 1).
digit(Latent) ~
    categorical([0, ..., 9], digit_classifier(Latent)).
vae_latent(Image, Component) ~
    normal(vae_encoder(Image, Component)).
vae_reconstruction(Latent) ~
    normal(vae_decoder(Latent), 0).
```

```
good_subtraction_image(Image1, Image2, Difference) :-
    vae_latent(Image1, ID) =:= prior1(ID),
    vae_latent(Image2, ID) =:= prior2(ID),
    LeftDigit is digit(vae_latent(Image1, logic)),
    RightDigit is digit(vae_latent(Image2, logic)),
    Difference is LeftDigit - RightDigit,
    concat(vae_latent(Image1, shape), LeftDigit, LeftEmb),
    concat(vae_latent(Image2, shape), RightDigit, RightEmb),
    reconstruction_loss(vae_reconstruction(LeftEmb), Image1),
    reconstruction_loss(vae_reconstruction(RightEmb), Image2).
```

Listing 4: Combining subtraction logic with a VAE in DeepSeaProbLog. The latent distribution for each of the two images is split into a `shape` and `logic` component. The `logic` component is regularised and used as input to the digit classifier `digit_classifier` while the `shape` component is only regularised and then attached to the most likely prediction of `digit_classifier`. This combination forms the input to the decoder, i.e., the decoder will generate an image of the attached digit. Note that the predicted digits also have to match the given subtraction result, which is how the digit classifier itself can be optimised.

**Parameters and neural architectures.** The NeSy VAE has two main neural components (Figure G.6), one for the VAE itself and another that handles the digit classification used in the subtraction logic. In contrast to the subtraction experiment, since the images are now separated, there is no need to determine the location of the digits. Similar to the subtraction experiment, a small set of 256 samples with direct supervision on the digit labels is again used to pre-train the classification portion of the overall network. All training utilised Adam as optimiser with a learning rate of $\cdot 10^{-3}$ and took 20 epochs using a batch size of 10. The pre-training was given 1 epoch with a batch size of 4.

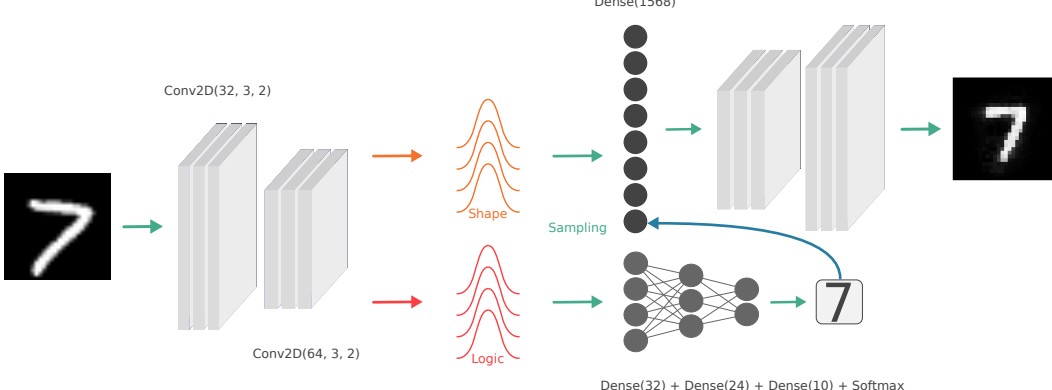

Figure G.6: VAE encoder-decoder architecture. The decoder is, apart from an initial dense layer, equal to the transpose of the encoder. Note how the latent distributions are split into a logic and shape component and the most likely prediction of the classifier is attached to the shape sample, which is then taken as input to a dense layer. All layers use ReLU activation functions, except the final convolutional one, which applies a hyperbolic tangent.

**Complications.** Regular VAE optimisation has two components: a Kullback-Leibler (KL) divergence term and a reconstruction loss term. Since DeepSeaProbLog requires probabilistic values, i.e., between 0 and 1, a probabilistic translation of these terms is necessary for optimisation in DeepSeaProbLog. The KL divergence term compares the latent distribution of the VAE to a standard normal prior and can as such be replaced by a $=:=$ comparison in the logic. The reconstruction loss is chosen to be the exponentiation of a negated average $L_1$ loss function, as it yields a value between 0 and 1 that can be interpreted as the probability that two images match. Specifically, the loss between

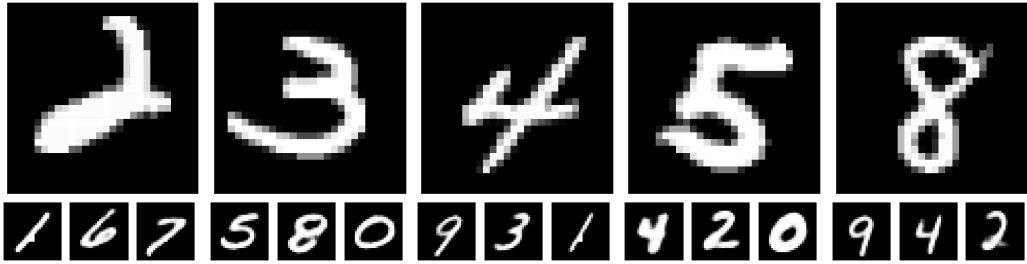

Figure G.7: Additional conditional query examples. 3 random difference values were given for 5 random minuends. All generated subtrahends were correct. Note the preservation of the style of the given minuends.

two such images $I_1, I_2 \in \mathbb{R}^{768}$ is given by

$$\exp\left(-\frac{1}{768}\sum_{i=1}^{768}|I_{1i} - I_{2i}|\right).$$ 

(G.1)

The latter can be interpreted as a form of soft unification (Rocktäschel and Riedel, 2017). While the usual function for soft unification is often chosen as a radial basis function, it is an average $L_1$ in our case. This loss function was chosen as it gave better and more crisp generations in comparison to a radial basis function.

**Additional results and interpretations.** Emphasis has to be put on the flexibility of generation in DeepSeaProbLog, as the generation of MNIST digits can be carried out in a range of different contexts without further optimisation. Indeed, one needs only construct a new predicate describing the logical context. The generative query that yields an image of a minuend and subtrahend that subtract to a given value is given in Listing 5. The conditional query that generates an image of a subtrahend given an image of a minuend and a difference value is given in Listing 6. Note in particular that the latter query generates subtrahends in the same 'style' as the given minuend, as can be visually confirmed by looking at the generations in Section 6.2. Additional conditional generations are given below in Figure G.7.

```
generate_subtraction(Difference, Generation1, Generation2) :-
    member(D1, [0, ..., 9]), member(D2, [0, ..., 9]),
    Difference is D1 - D2,
    attach(prior, D1, LeftEmb), attach(prior, D2, RightEmb),
    Generation1 is vae_reconstruction(LeftEmb),
    Generation2 is vae_reconstruction(RightEmb).
```

Listing 5: Given a difference value, generate images of a minuend and subtrahend that subtract to that value. The logic deduces all possible combinations for D1 and D2 that meet the subtraction evidence and attaches these to two random samples of the shape component. These two combinations lead to the two desired generations.

```
generate_conditional_subtraction(RightIm, Diff, LeftGen) :-
    member(D1, [0, ..., 9]),
    D2 is digit(vae_embedding(RightIm, logic)),
    Diff is D1 - D2,
    attach(vae_embedding(RightIm, shape), D1, LeftEmb),
    LeftGen is vae_reconstruction(LeftEmb).
```

Listing 6: Given an image of a subtrahend and a difference value, generate an image of a subtrahend. The subtrahend image is classified such that the logic can deduce the value of D1 that meets the given difference. By attaching that value of D1 to the shape component of the given subtrahend image, the VAE can generate an image of the correct minuend in the same 'style' of handwriting.

### G.3 Neural Hybrid Bayesian Network

**Setup details and DeepSeaProbLog program.** Our encoding of the neural hybrid Bayesian network of experiment 6.3 is given in Listing 7. The goal is to optimise the neural networks responsible for the classification of `humid` and `cloudy` conditions, as well as the network that predicts the temperature value. Additionally, we explicitly model the noise present on the true temperature labels as an learnable program parameter. To achieve this, a set of 1200 triples (`Im1`, `Im2`, `X`) is used as training set, where `Im1` is an MNIST digit of 0, 1 or 2 while `Im2` is an MNIST digit 8 or 9. In other words, we use MNIST digits as proxies for real imagery data. `X` is a set of 25 numerical meteorological features sampled from a publicly available Kaggle dataset (Cho et al., 2020). The label of each triple is the probability that the weather, as described by the correct labels of `humid`, `cloudy` and `temperature`, is good following our rules. Computing this probability label is non-trivial in and of itself. We utilised a large set of 1000 samples to approximate the correct underlying distributions and to obtain an approximate probability label.

```
humid(Im) ~ bernoulli(humid_detector(Im)).
cloudy(Im) ~ categorical(cloud_detector(Im), [0, 1, 2]).

temperature(X) ~ normal(temperature_detector(X), t(_)).
snowy_pleasant ~ beta(11, 7). rainy_pleasant ~ beta(1, 9)
cold_sunny_pleasant ~ beta(1, 1). warm_sunny_pleasant ~ beta(9, 2).

rainy(I1, I2) :-
    cloudy(I1) =\= 0, humid(I2) =:= 1.

good_weather(I1, I2, X) :-
    rainy(I1, I2) =:= 1, temperature(X) < 0, snowy_pleasant > 0.5.
good_weather(I1, I2, X) :-
    rainy(I1, I2) =:= 1, temperature(X) >= 0, rainy_pleasant > 0.5.
good_weather(I1, I2, X) :-
    rainy(I1, I2) =:= 0, temperature(X) > 15, warm_sunny_pleasant > 0.5.
good_weather(I1, I2, X) :-
    rainy(I1, I2) =:= 0, temperature(X) <= 15, cold_sunny_pleasant > 0.5.

P :: depressed(I1) :-
    cloudy(I1) =:= N, P is N * 0.2.

enjoy_weather(I1, I2, X) :-
    \+depressed(I1), good_weather(I1, I2, X).
```

Listing 7: The NDFs `humid(Im)` and `cloudy(Im)` classify a given sensory image as describing humid and cloudy conditions, respectively. `temperature(X)` takes a set of 25 numerical features `X` and predicts a mean temperature from those. Depending on the value of the temperature, 4 different cases of weather and their degree of pleasantness are described by beta distributions. We define `good_weather` as being true if the degree of pleasantness of any case is larger than 0.5. Finally, a person can be `depressed` with probability 0.2 or 0.4 depending on the degree of `cloudy`. Both then determine whether a person can enjoy the weather, if they are not `depressed` and `good_weather` is the case.

**Parameters and neural architectures.** We utilise a simple MNIST classifier (Figure H.8) in the NDFs `cloudy` and `humid`, while the network in the neural predicate `temperature` is a 3-layer, fully connected network with layers of size 35, 35 with ReLU activations and 1 with linear activation. Note that our classifiers share a common set of convolutional layers, requiring the learning of features that generalise to both classification problems. Additionally, the noise on the temperature prediction is modelled explicitly as a learnable TensorFlow variable with an initial value of 10. This choice is not arbitrary, as the initial neural parameter estimate will hover around the middle of the possible temperature values and a choice of 10 as initial standard deviation allows covering the entire range of temperature values with a non-insignificant probability mass. In this way, gradient information across the entire temperature domain can be accumulated during learning. Finally, DeepSeaProbLog was trained for 20 epochs using Adam with learning rate $10^{-3}$ and batch size of 10.

**Complications.** Ideally, simple 0-1 labels of `enjoy_weather` would be more intuitive, as we often do not observe the probability of an event but single cases where it is either true or false. However, our experiments have showed that our small dataset is insufficient to find an optimal solution using such labels in conjunction with the very distant supervision. To show that DeepSeaProbLog is still able to find solutions in cases where the supervision is slightly less distant using only 0-1 labels, we added a different neural hybrid Bayesian network experiment in Section H based on the famous burglary-alarm example of probabilistic logic (Listing 8).

**Additional results and interpretations.** We want to stress that learning to predict the right mean temperature from the distant supervision is not straightforward. The only learning signal for the temperature has to pass through PCFs with a very wide range, meaning they do not specify the exact temperature value immediately. Additionally, these PDFs still do not directly influence the supervision of `enjoy_weather`, only `good_weather`. The Gaussian noise that renders the temperature into a continuous random variable only further convolutes the task of determining the temperature. We conclude that DeepSeaProbLog can really extract meaningful learning signals from reasonably distant supervision.

## H ADDITIONAL EXPERIMENT

As briefly mentioned in the experimental section G.3, one more experiment was performed to show the promise of neural probabilistic logic programming in discrete-continuous domains in a more practical setting of 0-1 observations.

### H.1 NEURAL-CONTINUOUS BURGLARY ALARM

**Setup details and DeepSeaProbLog program.** The neural-continuous burglary alarm (Listing 9) extends the classic example from Bayesian network literature (Listing 8).

```
0.1 :: earthquake.
0.3 :: burglary.
0.9 :: hears.

0.7 :: alarm :- earthquake.
0.9 :: alarm :- burglary.

calls :- alarm, hears.
```

Listing 8: Classical burglary-alarm ProbLog program. Three probabilistic facts for the events `earthquake`, `burglary` and `hears` are given with their probabilities. The neighbour calls when hearing an alarm, while an alarm can go off because of an earthquake or a burglary.

Each data sample is a triple $(E, B, L)$, where $E$ can be an MNIST digit 0, 1 or 2 while $B$ can be an MNIST 8 or 9. Values for $E$ of 0, 1 and 2 correspond to no earthquake, a mild earthquake or a heavy earthquake respectively. If $B$ is an MNIST 8, then there is no burglary. If it is 9, then there is a burglary. $L$ can have either the value 0 or 1, indicating whether the neighbour called or not. Our dataset contains 12 000 such triples for training, while having 1 000 for validation and 2000 for testing purposes. Obtaining the weak supervision $L$ is done by taking the true probability of calling given the input and then randomly sampling according to that probability. To compute that true probability, a single sample is taken from the neighbour's true distribution. This true distribution has respective means of 6 and 3 for the horizontal and vertical Gaussian while both directions have a standard deviation of 3. Additionally, there are two possible ways to express that the distance of the neighbour should be smaller than 10 distance steps before hearing the alarm. One can use either the squared distance or the true distance in the rule `hears`. A separation is often maintained in the weighted model integration literature (Zuidberg Dos Martires et al., 2019) between comparison formulae that are polynomial and those that are generally non-polynomial. To illustrate that DeepSeaProbLog can deal with both classes of formulae, we will perform experiments for both the squared distance (polynomial, Listing 9) and the true distance (non-polynomial, Listing 10).

```
earthquake(Im) ~ categorical([0, 1, 2], earthquake_detector(Im)).
burglary(Im) ~ categorical([8, 9], burglary_detector(Im)).
```

```
neighbour_x ~ normal(t(μₓ), t(σₓ)).
neighbour_y ~ normal(t(μ_y), t(σ_y)).

hears :-
    neigbour_x * neighbour_x + neighbour_y * neighbour_y < 100.

P :: alarm(EarthquakeIm, _) :-
    earthquake(EarthquakeIm) =:= N, P is N * 0.35.
0.9 :: alarm(_, BurglaryIm) :-
    burglary(BurglaryIm) =:= 9.

calls(EarthquakeIm, BurglaryIm) :-
    alarm(EarthquakeIm, BurglaryIm), hears.
```

Listing 9: Our extension of the burglary alarm example has two neural detectors responsible for detecting earthquakes, `earthquake_detector`, and burglaries, `burglary_detector`. Additionally, whether or not the neighbour can hear the alarm if it goes off depends on the spatial distribution of this neighbour that is modelled as a two-dimensional Gaussian distribution. This distribution is randomly initialised and its parameters also need to be optimised. Note that `t(_)` is ProbLog notation for a single optimisable parameter. In DeepSeaProbLog, these are also considered to be within the set of neural parameters $\Lambda$.

```
hears :-
    sqrt(neighbour_x*neighbour_x + neighbour_y*neighbour_y) < 10.
```

Listing 10: Using the true distance in the `hears` predicate as a case of a non-polynomial comparison formula. Note that DeepSeaProbLog has support for advanced operators, such as `sqrt`.

**Parameters and neural architectures.** The complete neural architecture of both the earthquake and burglary classifiers is given in Figure H.8. Note that, even though their input consist of different sets of MNIST digits, we enforce a shared set of convolutional layers. In addition to the neural parameters in these networks, four independent parameters are present in the program. These are used as the means and standard deviations for the neighbour's spatial distribution and are randomly initialised. Specifically, the means are sampled uniformly from the interval $[0, 10]$ while the standard deviations were sampled from $[2, 10]$. All optimisation was performed using regular stochastic gradient descent with a learning rate of $8 \cdot 10^{-2}$ for two epochs using a batch size of 10.

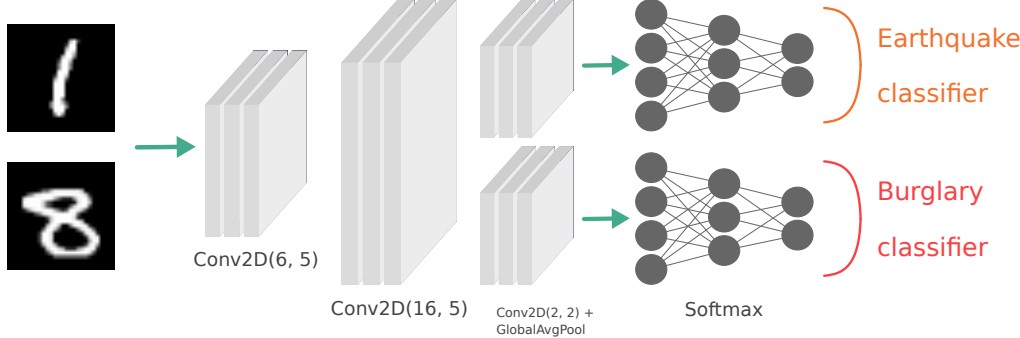

Figure H.8: Overview of the architecture of the earthquake and burglary networks. Both share two convolutional layers on top of which each then applies their own final convolutional layer followed by a global average-pooling operation that reduces the dimension to the number of classes. Finally, a softmax activation translates the output to probabilities. All other activation functions are ReLU functions.

**Complications.** Because of the difference in nature between the parameters in the neural networks and the four independent parameters in the Gaussian distribution, the latter required a boosted learning rate to provide consistent convergence. Specifically, the gradients for these four parameters were multiplied by a value of 20, which was found by a hyperparameter optimisation on the validation set.

**Results and interpretation.** Initial learning progress of the neural networks seems volatile (Figure H.9), which is likely due to the unoptimised state of the neighbour's spatial distribution. Two epochs of training proves to be sufficient to optimise both the neural detectors and the distribution. In fact, the earthquake and burglary classifiers converge to respective test accuracies of $98.73^{+0.22}_{-0.16}$ and $98.43^{+0.66}_{-0.50}$ when using the squared distance and very similar results for the true distance. The 4 parameters of the neighbour's distribution do not converge to the true values, but that is to be expected as they are underspecified. However, they do converge to values that provide satisfaction rates of the comparison formula in `hears` that are close to those of the true underlying distribution. All in all, three conclusions can be drawn. First, this experiment indicates that DeepSeaProbLog is capable of jointly optimising neural parameters and independent, distributional parameters. Second, DeepSeaProbLog seems to be able to fully exploit both polynomial and more general non-polynomial comparison formulae. This shows the strength of our approximate approach, as exact methods often fail to efficiently deal with non-polynomial formulae (Zuidberg Dos Martires et al., 2019). Third, DeepSeaProbLog can deduce meaningful probabilistic information from weak labels. Indeed, in order to optimise the neural detectors and the distribution, DeepSeaProbLog had to aggregate meaningful update signals from the 0-1 labels across the given training data set to approximate the underlying probability of `calls`.

To illustrate the strength of this final conclusion, consider the following. Assume that a burglary occurs and that the neural detector correctly classify this occurrence, then the absolute difference in $P(\texttt{alarm(EarthquakeIm, BurglaryIm)})$ between a mild `earthquake(EarthquakeIm) =:= 1` or a heavy earthquake `earthquake(EarthquakeIm) =:= 2` is only

$$|0.9 + 0.35 - 0.9 \cdot 0.35 - (0.9 + 0.7 - 0.9 \cdot 0.7)| = 0.035, \tag{H.1}$$

using Bayes' rule. Hence, a mild earthquake only has a very small effect on the overall probability, let alone in the case where the supervision itself is not even probabilistic.

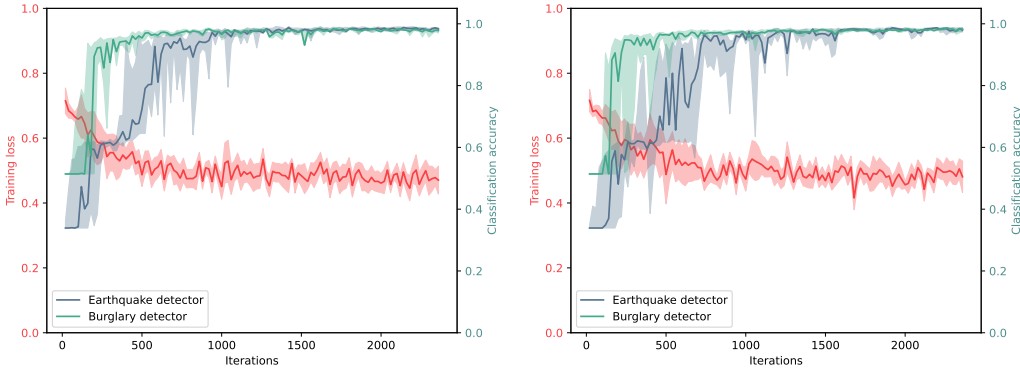

Figure H.9: Evolution of the training loss and validation accuracy of the neural 'earthquake' and 'burglary' detectors. For both squared (left) and true distance (right), the discrete supervision seems to be sufficient to facilitate meaningful learning.

## I LIMITATIONS

The main limitation of DeepSeaProbLog is one that it inherits from probabilistic logic in general, computational tractability. Efficiently representing a probabilistic logic program is done via knowledge compilation, which is $\#P$-hard. Once the probabilistic program is knowledge compiled, evaluating

the compiled structure is linear in the size of this structure. Inference remains linear in the size of the compiled structure after the addition of continuous random variables as all samples can be run in parallel with the current inference algorithm.

Although our sampling strategy is efficient in the sense that it is linear in the number of samples and it can be executed in parallel for each sample, it remains very simple. At its core, we utilise importance sampling to estimate the integration and this is known to not scale well to high dimensional spaces. More intricate inference strategies exist within the field of weighted model integration (Morettin et al., 2021), yet they currently lack the differentiability property to be integrated in DeepSeaProbLog's gradient-based optimisation. Conversely, our examples illustrate that our naive strategy is sufficient to solve basic tasks. Moreover, more intricate sampling strategies do not always scale well under logical or algebraic constraints and so importance sampling techniques are still considered state-of-the-art (Nitti et al., 2016; Tolpin et al., 2016). It is still an open question how to perform successful joint inference and gradient-based learning under such constraints.

Orthogonal to the estimation of the integral during inference, exact knowledge compilation also prevents the scaling of DeepSeaProbLog to larger problem instances. Approximate knowledge compilation is the field of research that deals with tackling this issue. While it contains interesting recent work (Fierens et al., 2015; Huang et al., 2021; Manhaeve et al., 2021b), it was highlighted by Manhaeve et al. that the introduction of the neural paradigm does lead to further complications. As such, we opted for exact knowledge compilation, but it has to be noted that we will be able to benefit from any future advances in the field of approximate inference. Alternatively, different semantics (Winters et al., 2022) can simplify inference, but they lead to a degradation of expressivity of the language.

A potential future avenue for scaling up DeepSeaProbLog inference would be the use of further continuous relaxation schemes. More specifically, replacing discrete random variables with relaxed categorical variables (Maddison et al., 2017; Jang et al., 2017) might allow us, for instance, to forego the knowledge compilation step while still being able to pass around training signals.

