# OpenReview forum: "Neural Probabilistic Logic Programming in Discrete-Continuous Domains"
_ICLR.cc/2023/Conference — Submitted to ICLR 2023_

### Official Review · Reviewer_6UoH · 2022-10-27

**Confidence:** 4
**Correctness:** 4
**Technical Novelty And Significance:** 3
**Empirical Novelty And Significance:** 2
**Recommendation:** 5

**Clarity, Quality, Novelty And Reproducibility:**

 The work is fairly original if a bit incremental. The paper would greatly benefit from
 better organization of the material so that more of the novelty can move from the
 appendix back into the main text. I would have preferred more details on the semantics
 of the language than the brief sections included. The work while ostensibly isn't
 very reproducible is effectively describing a system which I expect to be publicly
 released.

Update:

I thank the authors for their response, but I still feel the clarity of the work could be greatly improved. If this paper is accepted, it would greatly benefit from more detail around the method itself. How is differentiation through WMI performed? Does knowledge compilation mean we don't have to worry about differentiating through discrete variables?

**Strength And Weaknesses:**

The paper continues exploring the very interesting intersection of
deep learning and logic programming for solving problems each set of
approaches would struggle on their own.

Unfortunately, the paper feels far too incremental. Even the example seems fairly
similar to the example used in Raedt et al's 2019 IJCAI workshop submission. With
both examples feeling fairly contrived. The neural baseline seems unusually weak.
I would expect a more reasonable architecture to use a Spatial Transformer Network
as those have been a mainstay of object localization for quite a while.

I have some frustrations with large portions of the main text being essentially
background information on logic programming. This lead to much of the unique
contributions of the paper being relegated to the appendix. While I understand the
need to explain logic programming to many of ICLR typical readers, I would have
still preferred a far more condensed background section.

This especially lead to issues as its not clear the paper really introduced the
semantics of DeepSeaProbLog. There were some definitions of how to calculate the probability
of a possible world and a query, but further details are deferred to another paper.
As one of the major contributions of the paper more details should have stayed
in the main text.

 Questions:

 What is the semantics of `normal` in Example 3.1? I expect normal to take two arguments
 but it only takes one. Does it's argument return a tuple?

 Minor:

 Not all code examples and figures have labels


**Summary Of The Paper:**

This paper introduces the neural probabilistic logic programming language DeepSeaProbLog, which
can be seen as an extension of DeepProbLog. To DeepProbLog it adds the ability to represent
and use probability distributions of continuous and mixed continuous and discrete spaces.
The paper after describing the language provides an illustrative example of a classification
task which a neural baseline struggles on, but DeepSeaProbLog with its reasoning capabilities
excels at.


**Summary Of The Review:**

Interesting work that would benefit from more clarity or an experiment that feels less contrived.

---

> ### Author Response · Authors · 2022-11-09
> **Addressing of concerns/questions**
>
> 1. We would argue DeepSeaProbLog goes beyond a simple incremental extension of DeepProbLog, as addressed in the general comment to all reviewers.
>
> 2. With respect to the neural baseline, the point of the experiment is not to achieve the absolute best performance. Rather, we showcase how DeepSeaProbLog can augment any neural network architecture with discrete-continuous background knowledge and that doing so yields the expected improvements in data-efficiency and generalisation performance.
>
> 3. We would like to clarify that the only component with background knowledge is Section 2. Section 3 immediately introduces novel concepts such as the neural distributional fact (Definition 3.2) and probabilistic comparison formulas (Definition 3.3) that are fundamental building blocks of a DeepSeaProbLog program.
>
> 4. The semantics of a probabilistic logic programming language are uniquely determined by the probability of a possible world, which itself is given meaning by the variables that are declaratively defined by the neural distributional facts. The probability of a possible world then determines the probability of a query, by aggregating the probability of each possible world that makes the query true. In other words, the semantics are uniquely determined by our given definitions.
>
> Question:
>
> “normal” should be, according to definition 3.1, interpreted as a probability distribution. In this case that distribution is a Gaussian, or normal distribution that indeed takes 2 arguments. The name chosen to denote this distribution is a purely syntactic matter, not one of semantics.
> As mentioned in Example 3.2, the argument “temperature_predictor(Data)” indeed returns a tuple containing the mean and standard deviation parameters required by the normal distribution.

---

### Official Review · Reviewer_ViX9 · 2022-10-28

**Confidence:** 3
**Correctness:** 2
**Technical Novelty And Significance:** 2
**Empirical Novelty And Significance:** 2
**Recommendation:** 5

**Clarity, Quality, Novelty And Reproducibility:**

The authors don't give a good explanation of how they are handling discrete random variables despite claiming this in the abstract. There is a brief mention of using Dirac Delta distributions but more details would be needed to understand how they plan to reparameterize these Dirac Delta distributions. Categorical random variables that can take on multiple unordered values need more thought. I would have liked to see some experiments involving such variables.

Also please note that integrating out discrete random variables doesn't count as supporting discrete random variables in the language.

Weighted model counting is a well known technique. It's unclear to this reviewer whether differentiation over weighted model counting by using relaxation of indicators counts as an innovation.

A lot of the algorithms here such as the sum-product terms are referring to other papers which make it somewhat harder to reproduce.

I had some difficulty following the subset of horn clauses that are supported or whether concepts like unification in Prolog are included. The example program on page 3 included a term "Degree" which was not used in the query. This needs some clarity is that being integrated out?

The example of digit differencing didn't seem very impressive for a language that is claiming to combine logic, probability and neural networks.

**Strength And Weaknesses:**

The strengths of the paper are they are putting together various well-studied concepts and techniques into one paper. However, this is also a weakness because it is hard to distinguish the key contributions of the authors. The techniques of using reparameterization of continuous random variables or relaxations are well known and the probabilistic logic program's semantics also seem to not be novel. So it is not obvious as to what is the key innovation here.



**Summary Of The Paper:**

The paper presents a probabilistic programming language based on extending a prolog-like language with probabilistic facts and equipping a query with probabilistic semantics. The language includes both continuous and discrete random variables and it includes parameterized neural networks.

The authors describe a fairly elaborate learning technique whereby gradients can be computed of the weighted model counting by using reparameterization of continuous random variables and relaxation of indicators. This allows them to train the neural network parameters.

The authors then show a couple of results demonstrating how a combination of probabilistic logic programs and neural networks  can be used to achieve some interesting results including computing the numerical difference of two digits shown pictorially as well as a variational encoder that reverses this.

**Summary Of The Review:**

The description of their work is reasonable but it is not clear what is the key novelty is among the hodge lodge of ideas that are included in this paper besides how do they even support discrete random variables is not well covered.

---

> ### Author Response · Authors · 2022-11-09
> **Addressing of comments**
>
> 1. We address the incrementality concern in our general comment by explicitly restating our contributions.
>
> 2. Inference for random variables over finite sample spaces are handled in exactly the same fashion as in DeepProbLog, including categorical random variables, using exact, symbolic methods. This is a standard, well-studied and differentiable inference strategy in probabilistic logic programming.
> We do have experiments that involve such variables, such as “digit” in the MNIST subtraction (6.1) and “cloudy” in the neural hybrid Bayesian network (6.3).
>
> 3. The choice of expression 'marginalised out' was an unfortunate choice of words and does not represent what actually happens. Instead, we should have used the term 'perform exact symbolic inference', which has been altered in the text.
>
> 4. We use a generalisation of weighted model counting, called weighted model integration (WMI), since weighted model counting only applies to discrete domains. Differentiable WMI has as of yet not been a focus of the WMI literature and our approach is hence the first to propose a practical implementation that does support differentiability. See also the general comment for more information on this differentiability contribution.
>
> 5. Similar to ProbLog (and DeepProblog) the semantics for DeepSeaProbLog are only defined for so-called sound programs [1] (this includes unification). We have added this information to the paper at the end of Appendix B.
>
> 6. The absence of a value for the term Degree was a typo and has been fixed.
>
> 7. While making use of a rather simple dataset, it is already sufficient to showcase that DeepSeaProbLog goes beyond what existing methods can do with respect to flexibility and data efficiency.
> Larger or more complicated neural networks might be able to achieve higher performance, but they will require inordinately more data and computation power. Additionally, because we subsume the neural paradigm, any better-performing network can always be further augmented with discrete-continuous background knowledge in DeepSeaProbLog.
> Looking at fellow neural-symbolic approaches, they can only use point estimates and can’t take into account the full support of a distribution which leads to lower performance in general (Table 1, 6.1). Moreover, they lack the probabilistic semantics to perform generative tasks (NeSy VAE, 6.2).
>
>
> [1] Riguzzi, Fabrizio, and Terrance Swift. "Well–definedness and efficient inference for probabilistic logic programming under the distribution semantics." Theory and practice of logic programming 13.2 (2013): 279-302.

---

### Official Review · Reviewer_tpMA · 2022-10-29

**Confidence:** 4
**Correctness:** 4
**Technical Novelty And Significance:** 2
**Empirical Novelty And Significance:** 2
**Recommendation:** 5

**Clarity, Quality, Novelty And Reproducibility:**

Clarity: good
Quality: good
Novelty: it looks like an increment on top of DeepProblog
Reproducibility: authors provide material in the appendice that may be sufficient for reproducibility of experiments, but I haven't tested.

**Strength And Weaknesses:**

The work looks quite interesting and fills a gap related with the
lack of probabilistic logic programming to handle continuous
variables. On the other hand, it looks like a small increment related
with DeepProblog.

It would be interesting to add more applications in the experimental
section. Also, what is the scalability of the implementation?

**Summary Of The Paper:**

This paper describes DeepSeaProblog, a neuro-symbolic probabilistic
logic programming language and system that can handle continuous
variables.

**Summary Of The Review:**

This paper describes DeepSeaProblog, a neuro-symoblic probabilistic
logic programming language and system that can handle continuous
variables.

The work looks quite interesting and fills a gap related with the
lack of probabilistic logic programming to handle continuous
variables. On the other hand, it looks like a small increment related
with DeepProblog.

It would be interesting to add more applications in the experimental
section. Also, what is the scalability of the implementation?

Obs:
COPPE Gerson Zaverucha in Garcez 2022 reference should be Gerson
Zaverucha. COPPE is the institution.

---

> ### Author Response · Authors · 2022-11-09
> **Addressing of comments**
>
> We address the incrementality concern in our general comment by explicitly restating our contributions.
>
> Concerning the scalability of our implementation, we note that problems expressible in DeepSeaProbLog fall into the #P-hard complexity class, even when sampling continuous random variables (Appendix H). This means that in the general case problems are computationally hard to solve. Recent work in the NeSy community has looked at circumventing this computational hardness by using approximations [1,2] or restricting the problem class [3]. This is an interesting direction for future work in the context of DeepSeaProbLog.
>
> [1] Huang, J., Li, Z., Chen, B., Samel, K., Naik, M., Song, L., & Si, X. (2021). Scallop: From probabilistic deductive databases to scalable differentiable reasoning. Advances in Neural Information Processing Systems, 34, 25134-25145.
> [2] Manhaeve, R., Marra, G., & De Raedt, L. (2021). Approximate Inference for Neural Probabilistic Logic Programming. In KR (pp. 475-486).
> [3] Winters, T., Marra, G., Manhaeve, R., & De Raedt, L. (2022, June). Deepstochlog: Neural stochastic logic programming. In Proceedings of the AAAI Conference on Artificial Intelligence (Vol. 36, No. 9, pp. 10090-10100).

---

> ### Comment · Reviewer_tpMA · 2022-12-01
> **Regarding more benchmarks or applications**
>
> Why do you apply your approach to only two toy applications? I appreciate the work, but it'd be nice to have an idea of how would the model behave on other benchmarks/applications. For example, would it be possible to use data from Tensorflow Probability? (some are available from kaggle).

---

> > ### Author Response · Authors · 2022-12-02
> > **Concerning applications**
> >
> > We are not sure what data the reviewer is referring to, but from an online search we encountered the following Kaggle example [1]. Here TensorFlow probability is used to encode Bayesian neural networks (BNN). That is, neural networks whose parameters are distributed according to certain distributions.
> >
> > This is different to the question we are tackling, where we use deterministic neural networks to parametrize probabilistic logic programs. It is indeed an interesting question, for future research, whether it makes sense to use BNNs instead of deterministic NNs in a neural-symbolic programming setting.
> >
> > Note that Experiment 6.3 uses real meteorological data [2] available from Kaggle [3], which we mentioned in the Appendix, and deterministic NNs to obtain the parameters of the normally distributed temperature variable used in the logic.
> >
> > [1] https://www.kaggle.com/code/usharengaraju/tensorflow-probability-probabilisticbnn
> >
> > [2] Dongjin Cho, Cheolhee Yoo, Jungho Im, and Dong-Hyun Cha. Comparative assessment of various machine learning-based bias correction methods for numerical weather prediction model forecasts of extreme air temperatures in urban areas. Earth and Space Science, 7(4):e2019EA000740, 2020.
> >
> > [3] https://www.kaggle.com/datasets/viktorpopov/bias-correction-ucl

---

> > > ### Comment · Reviewer_tpMA · 2022-12-09
> > > **Many thanks for your clarification**
> > >
> > > Many thanks for the clarifications. It helped me to understand better what is being proposed.

---

### Official Review · Reviewer_qtFY · 2022-10-30

**Confidence:** 4
**Correctness:** 4
**Technical Novelty And Significance:** 3
**Empirical Novelty And Significance:** 3
**Recommendation:** 8

**Clarity, Quality, Novelty And Reproducibility:**

The paper is well-written, and the running examples are nice.

The technical part of the main paper is sound, though I did not check all details and the proofs in the appendix. In terms of empirical evaluation, there is one minor weakness: some auxiliary tasks are provided in the MNIST subtraction experiment. It is unclear how these tasks affect the performance of the baselines. Please refer to the detailed comments above.

The main novelty of this paper is to extend DeepProbLog to continuous domains.

**Strength And Weaknesses:**

The main contribution of DeepSeaProbLog is to provide a more general language for formulating NeSy rules/programs. This naturally leads to broader applications of NeSy methods. However, one potential downside is that approximations need to be applied to ensure the algorithm's efficiency.

- In terms of computing WMC/WMI, DeepProbLog uses a backward search-based exact solver while DeapSeaProbLog approximates the result by sampling, even if all variables in the probabilistic program are boolean. In this case, it would be nice to see the performance difference between the two methods. Efficiency comparison between (i) the compilation phase of DeepProbLog, (ii) the execution phase of DeepProbLog (evaluating the compiled logic circuit), and (iii) the execution phase of DeepSeaProbLog would also provide a better overview of the tradeoff between efficiency and performance.

- Following the above comment, is it possible to combine knowledge compilation techniques with sampling to get the best of both worlds? That is, apply knowledge compilation to parts that can be efficiently turned into ``recursively nested sums of products'', and sample the part with complex variable dependencies.

- In the MNIST subtraction experiment, some samples provided location supervision and curriculum learning. The authors mentioned that this is necessary for DeepSeaProbLog since otherwise it may converge to trivial cases. If this phenomenon also happens in baseline methods?

- If a PCF contains only one variable, which is often the case in the probabilistic programs adopted in the paper, it seems that we can "define" this PCF as a boolean variable and simplifies the computation of the WMI by computing the probability of this PCF from the NN. Will this be able to achieve a better performance-efficiency tradeoff?

**Summary Of The Paper:**

This paper proposes DeepSeaProbLog, a NeSy algorithm that supports rules and facts specified in both discrete and continuous domains. DeepSeaProbLog falls into the category of NeSy method that injects (logical) constraints in neural networks. The main contribution of this paper is the generalization of supported logical programs from boolean variables to continuous variables (e.g., Gaussians). The resultant probabilistic program becomes an SMT formula. Since exact probabilistic inference over SMT formulas is intractable, the paper adopts several well-established methods to approximate the queried probability. The authors introduce a new task, i.e., MNIST subtraction, that requires algorithms to simultaneously learn the bounding boxes of MNIST digits and compute their subtracted values. On this task and two other existing NeSy tasks, DeepSeaProbLog performs better than neural networks and some other NeSy baselines.

**Summary Of The Review:**

In summary, I vote for acceptance of the paper as it proposes a NeSy algorithm that supports rules and facts specified in both discrete and continuous domains. The main weakness of the paper is the insufficient discussion of the performance-efficiency tradeoff, as detailed in the comments above.

---

> ### Author Response · Authors · 2022-11-09
> **Addressing of comments**
>
> 1. DeepSeaProbLog, by default, only applies approximations and sampling for random variables with infinite sample spaces. The other ones are still dealt with in an exact, symbolic way. Hence, if all variables in the program have finite sample spaces, DeepSeaProbLog inference is, by default, exactly equal to DeepProbLog inference (see general comment).
>
> 2. Does the comment made above answer the question? DeepSeaProbLog’s inference based on Sampo already combines knowledge compilation with sampling in a differentiable fashion.
>
> 3. This phenomenon indeed also occurs in the two other approaches. Without any direct location supervision, the disconnect between classification and regression in two-stage object detectors will always prevent successful learning. Hence, the same samples with additional location supervision were also given to the other approaches in the same, continual learning memory fashion.
>
> 4. It is an interesting observation, which corresponds to pushing the integration into the sum-product structure to the level of the leaves. This is already done for the variables with finite sample spaces (see for example how we push the integral to the PCF in Equation A.8 in the proof of Proposition A.1). If we were to apply such a pushing down strategy in a more general setting we would need to adapt techniques already present in the WMI literature and this could indeed reduce the variance of Monte Carlo estimates.

---

### Official Review · Reviewer_zi3P · 2022-10-31

**Confidence:** 4
**Correctness:** 4
**Technical Novelty And Significance:** 4
**Empirical Novelty And Significance:** 4
**Recommendation:** 6

**Clarity, Quality, Novelty And Reproducibility:**

This work is overall well-written and the contribution is solid and novel to me.

**Strength And Weaknesses:**

The motivation of this work to enable neural probabilistic programming to work in mixed discrete-continuous domains is tempting since it would allow for expressive modeling for real-world problems. The use of weighted model integration tool is novel to me and it is the key to tackle the mixed discrete-continuous domain challenge. The connection between the proposed DeepSeaProbLog and the existing work on neural probabilistic programming is nicely explained. Still, here are some of my concerns/suggestions:

- Missing references to some of the current literature on weighted model integration solvers such as [1,2,3]. I think this work would benefit from a discussion on the choice of WMI solvers for performing inference in DeepSeaProbLog. For example, how different WMI solvers would support different inference performances of DeepSeaProbLog.
- The proof of Prop 4.1 refers to Zuidberg Dos Martires et al. (2019) while it is unclear which results in Zuidberg Dos Martires et al. (2019) is related to the conclusion that Eq C.3 is indeed a weighted model integration problem.
- Missing comparison in the pure discrete setting. When DeepSeaProbLog is applied to a pure discrete setting, there should be a bunch of neural probabilistic programming benchmarks as well as baselines for comparison. The authors might want to put such an empirical comparison to illustrate the discrete reasoning capability of DeepSeaProbLog in such settings.
- In Sec 3.2, it seems that one limitation of DeepSeaProbLog is that each distributional fact must define a different random variable. I wonder why such an assumption is necessary. Also, are there any distributional assumptions on the continuous variables? It seems that the continuous variables are all assumed to be Gaussian.
- In the neural-symbolic VAE experiment, it would be more convincing to include an ablation study where the VAE has no difference constraint but is still trained with difference as addition input. This ablation study is necessary since it might be possible that the VAE might simply learn the digit pair conditioned on the difference label and such an ablation study would help to see how much the DeepSeaProbLog help improve accuracy.
- Another issue in the neural-symbolic VAE experiment is that when it shows that DeepSeaProbLog is able to answer conditional generative queries, only one example is presented. This can be further improved by presenting some metrics such as accuracy to measure the performance of answering such queries.

[1] P. Morettin, A. Passerini, and R. Sebastiani. Efficient weighted model integration via SMT-based predicate abstraction. In IJCAI, 2017.
[2] Z. Zeng, P. Morettin, F. Yan, A. Vergari,
and G. Van den Broeck. Probabilistic inference with algebraic constraints: Theoretical limits and practical approximations. In NeurIPS, 2020.
[3] Z. Zeng, P. Morettin, F. Yan, A. Vergari,
and G. Van den Broeck. Scaling up hybrid probabilistic
inference with logical and arithmetic constraints via message passing. In ICML, 2020.

**Summary Of The Paper:**

This work proposes a neural probabilistic programming language that supports both discrete and continuous variables, called DeepSeaProbLog. An implementation of DeepSeaProbLog that allows inference and gradient-based learning is further proposed, by leveraging a reduction to weighted model integration and differentiation through a weighted model integral. Empirical evaluations on DeepSeaProbLog are presented on a neural-symbolic object detection task, variational auto-encoder with a difference constraint in the latent space, and neural hybrid Bayesian networks.

**Summary Of The Review:**

The proposed DeepSeaProbLog is novel to me. However, my main concern is the empirical evaluation not being extensive and not so convincing.

---

> ### Author Response · Authors · 2022-11-09
> **Addressing of concerns/suggestions**
>
> 1. While different WMI solvers could be used to tackle the inference component of DeepSeaProbLog, our choice was made out of a need for differentiability. While many WMI solvers are able to exploit complex logical or algebraic dependencies, the constructions that facilitate such an exploitation are non-differentiable and do not admit any immediate differentiable approximation. Such a solver, can hence, not be used in our setting where we are interested in the gradient computations for all parameters of a hybrid probabilistic logic program.
>
> 2. We have added the following sentence at the very end of Appendix C:
> Specifically, line P2 in the proof of Theorem 2 in Zuidberg Dos Martires et al. (2019) corresponds to C.3, which is shown to be equal to an instance of WMI.
>
> 3. In the purely discrete setting, DeepSeaProbLog inference corresponds exactly to DeepProbLog inference, hence any purely discrete comparison would not contribute any additional information (see general comment).
>
> 4. Having more than one distribution for the same random variable would not make sense on a semantics level. For instance, if we had the distributional facts
>
> x ~ normal(20, 3). x ~ beta(1, 1).
>
> This would mean that x were to be distributed according to two different distributions at the same time, for which we do not have an interpretation.
>
> As for the assumptions on continuous random variables, we require that they follow a distribution that admits a reparametrisation. In practice, this encompasses all differentiable distributions offered by TensorFlow Probability, which are contained in the exhaustive list in the link below. Note that this includes multivariate variables.
>
> https://www.tensorflow.org/probability/api_docs/python/tfp/distributions
>
> Regarding only using Gaussian distributions: the MNIST subtraction experiment uses generalised normal distributions, while the neural hybrid Bayesian network experiment has beta-distributed variables, all of which can be differentiated through.
>
> 5. A conditional VAE trained on pairs of digits as proposed by the reviewer that uses the difference label as a conditioning factor would indeed be able to generate pairs of digits that satisfy the condition. However, such an architecture would not allow us to alter, in a zero-shot way, the query. The experiment in Section 6.2 shows that this is possible to achieve with DeepSeaProblog, by logically decomposing the generative distribution, where we train on generating valid minuends and subtrahends given the difference, and query on generating a valid subtrahend given the minuend and the difference.
>
> 6. We are not sure which metric the reviewer refers to as a VAE is usually not evaluated using accuracy. However, we acknowledge that showing only a single generated example seems not convincing. Therefore we added further generated examples to Appendix G.2 of the paper.

---

### Author Response · Authors · 2022-11-09
**Discrete variables, incrementality concerns and implementation release**

From the comments of the reviewers, we have decided to add the following paragraph to the main body of the paper to improve the discussion about how we handle discrete random variables. Additionally, we have added to the Appendix a figure showing how this symbolic inference intuitively works, as we can not go into details of this existing work in the main body of the paper (end Section 4.1).

“Note that the Sampo algorithm only samples random variables whose expected value with respect to the function SP(x) cannot be computed exactly. Hence, DeepSeaProbLog is able to perform exact symbolic inference for random variables with finite sample spaces, e.g., Boolean random variables. In turn, this means that in the absence of random variables with infinite sample spaces an implementation of DeepSeaProbLog using Sampo coincides with DeepProbLog on a semantics level (Proposition A.1) as well as on an inference level. In Appendix D we provide a diagrammatic representation of the function SP(x) for the query in Example 3.1 where we also perform exact symbolic inference for the discrete variable.”


We would also like to address the incrementality concerns of some reviewers by clearly listing our most significant contributions:

1. We link the semantics of a hybrid probabilistic programming language with support for neurally parametrised distributions to WMI inference and prove the correctness of this relation in Proposition 4.1.

2. We propose the first practical approach to rendering WMI inference differentiable, a topic which has of yet not been properly addressed within the WMI literature. We also prove that this approach, utilising continuous relaxations, results in unbiased derivative estimates in the infinite coolness limit (Proposition 4.2).

3. The proof of Proposition 4.2 also extends the result of Petersen et al. (2021), stating that relaxed functions coincide with their non-relaxed version in the infinite coolness limit, to the derivatives of relaxed functions. This is a novel theoretical contribution that goes beyond our specific probabilistic programming setting.

4. Apart from these theoretical contributions, we also provide an implementation of the theory that effectively merges DeepProbLog with TensorFlow Probability, bridging the gap between discrete, symbolic methods and continuous, approximative ones in a declarative framework. Additionally, our experiments show the early promise of this framework.


Finally, we want to note that we will publicly release our implementation of DeepSeaProbLog together with our full experimental setup upon acceptance.

---

### Decision · Program_Chairs · 2023-01-20

**Decision:**

Reject

**Justification For Why Not Higher Score:**

The presentation of the paper can be greatly improved. Many modeling choices are not contextualized and it is not clear why the proposed experiments are challenging.

**Justification For Why Not Lower Score:**

N/A

**Metareview: Summary, Strengths And Weaknesses:**

The paper deals with the important problem of performing inference in neuro-symbolic setting when in the presence of both continuous and discrete variables over which logical and arithmetic constraints are defined. The authors propose an extension of DeepProblog that incorporates a simple inference scheme to deal with mixed continuous-discrete variables, leveraging the framework of weighted model integration (WMI). Specifically, they use a form of vanilla Monte Carlo estimation via sampling in combination with relaxing hard constraints by introducing temperated sigmoid units.

The reviewers agreed that the paper addresses an important topic: how to integrate probabilities and constraints over mixed domains to reason with deep learning models. At the same time they highlighted several important shortcomings.

First, presentation can be greatly improved. This requires better explaining the role and interaction of continuous and discrete variables during learning and inference. E.g., while motivating and introducing WMI, resorting to delta of Dirac functions to deal with discrete variables is not needed. A simpler fix in presentation is to adopt the reduction from continuous to discrete introduced in [1]. This basically would boil down to clearly state in the motivation that one wants to extend DeepProblog to deal with continuous variables and arithmetic constraints over them. The challenges in inference and learning can be better explained for both the probabilistic programming community (who are dealing with continuous and discrete variables but not constraints) and the logic community (who knows how to deal with logical constraints via knowledge compilation).

Second, the approximations involved to backpropagate through discrete variables (reparametrization trick), approximate integrals (MC sampling) and softening hard constraints via tempered sigmoids are known since quite some time in the probabilistic programming literature. While reviewers agreed that an initial work might propose a simple inference scheme to begin tackling an important problem, they also recommend authors to better discussing what are the limitations of the combinations of these choices. I.e., the effect of these approximations is not discussed in practice. E.g., the reparametrization trick is known not to scale when several discrete distributions jointly (or even better sequentially) appear in a computational graph. Authors could discuss the effect of replacing these simple estimators with more recent gradient estimators with lower variance, e.g., [2]. Furthermore, annealing the temperature parameter is highly tricky.

Third, experiment-wise, there is a lack of proper baselines and details. As the classical tricks from the probabilistic programming literature have been used to approximate an intractable inference problem, authors can better motivate why are they not used in the comparisons. I believe some partial reason might lie in the specific modeling choice of parameterizing literals or using knowledge compilation to deal with logical constraints. However, reviewers highlighted how one could, in principle, use the same idea to brute force ground a constraint and marginalize discrete variables in other ppls, as the logical constraints in the shown experiments are of modest size. Lastly, to strengthen the idea of combining neuro ppls with linear constraints, authors could tackle experiments where linear constraints over continuous variables are crucial.

I personally believe that the paper can be very strong if the authors take into consideration all the reviewers' comment in a future resubmission. As it is, it is close but not yet ready for publication.

[1] - Zeng et al. "Efficient Search-Based Weighted Model Integration", 2020
[2] - Niepert et al. "Implicit MLE: Backpropagating Through Discrete Exponential Family Distributions" 2021

**Summary Of Ac-Reviewer Meeting:**

The meeting was very productive as it allowed reviewers to better understand the paper contributions and calibrate their evaluation.
At the same time. more concerns regarding the technical contribution in terms of novelty of inference emerged as well as some skepticism about the choice of the experiments chosen to highlight inference in mixed continuous-discrete domains.